# BiLO: Bilevel Local Operator Learning for PDE inverse problems

## Abstract

We propose a new neural network based method for solving inverse problems for partial differential equations (PDEs) by formulating the PDE inverse problem as a bilevel optimization problem. At the upper level, we minimize the data loss with respect to the PDE parameters. At the lower level, we train a neural network to locally approximate the PDE solution operator in the neighborhood of a given set of PDE parameters, which enables an accurate approximation of the descent direction for the upper level optimization problem. The lower level loss function includes the L2 norms of both the residual and its derivative with respect to the PDE parameters. We apply gradient descent simultaneously on both the upper and lower level optimization problems, leading to an effective and fast algorithm. The method, which we refer to as BiLO (Bilevel Local Operator learning), is also able to efficiently infer unknown functions in the PDEs through the introduction of an auxiliary variable. Through extensive experiments over multiple PDE systems, we demonstrate that our method enforces strong PDE constraints, is robust to sparse and noisy data, and eliminates the need to balance the residual and the data loss, which is inherent to the soft PDE constraints in many existing methods.

## 1 Introduction

A fundamental task across various scientific and engineering fields is to infer the unknown parameters of a partial differential equation (PDE) from observed data. Applications include seismic imaging (Deng et al., 2023; Martin et al., 2012; Yang et al., 2021b), electrical impedance tomography (Uhlmann, 2009; Molinaro et al., 2023), personalized medicine (Lipková et al., 2019; Zhang et al., 2024a; Schäfer et al., 2021; Subramanian et al., 2023), and climate modeling (Sen & Stoffa, 2013). PDE inverse problems are commonly addressed within the frameworks of PDE-constrained optimization (Hinze et al., 2008) or Bayesian inference (Stuart, 2010). In the PDE constrained optimization framework, the objective is to minimize the difference between the observed data and the PDE solution, and the PDE is enforced as a constraint using adjoint or deep learning methods. In the Bayesian inference framework, the inverse problem is formulated as a statistical inference problem, where the goal is to estimate the posterior distribution of the parameters given the data. This requires sampling parameter space and solving the forward PDE multiple times. Here, we develop a constrained optimization framework for solving PDE inverse problems using deep learning.

### 1.1 Related work

The **Adjoint Method** is a widely used technique for computing the gradients of the objective function with respect to the PDE parameters using numerical PDE solvers in the PDE-constrained optimization framework. This method provides accurate gradients and strongly satisfies the PDE constraint. However, the method requires explicitly deriving the adjoint equation and solving both forward and adjoint equations at each iteration, which is complex and computationally expensive, especially for nonlinear or high-dimensional problems (Hinze et al., 2008; Plessix, 2006).

**Physics-Informed Neural Networks (PINNs)** have emerged as novel methods for solving inverse problems in a PDE constrained optimization framework (Karniadakis et al., 2021; Raissi et al., 2019; Jagtap et al., 2022b;a; Chen et al., 2020; Zhang et al., 2024a; Yang et al., 2021a; Kapoor et al., 2024; Chen et al., 2020; Jagtap et al., 2022a; Zhang et al., 2024a). PINNs represent PDE solutions using

neural networks and embed both the data and the PDE into the loss function through a mesh-free approach. By minimizing the total loss, PINNs effectively solve the PDE, fit the data, and infer the parameters simultaneously, showcasing integration of mathematical models with data-driven learning processes. A related approach, **Optimizing a Discrete Loss (ODIL)**, utilizes conventional numerical discretizations of the PDEs and the loss is minimized over the parameters and the PDE solutions at the grid points rather than the weights of a neural network (Karnakov et al., 2022; Balcerak et al., 2024). However, in these methods, the PDE is enforced as a soft constraint, which requires balancing the residual and the data loss, and can lead to a trade-off between fitting the data and solving the PDE accurately.

**Neural Operators (NOs)** aim to approximate the PDE solution operator (parameter-to-solution map) and can serve as surrogate models for the forward PDE solvers (Kovachki et al., 2022). Once these surrogates are established, they can be integrated into a Bayesian inference framework or other optimization algorithms to solve inverse problems, leveraging the speed of evaluating a neural network (Zhou et al., 2024; Pathak et al., 2022; Lu et al., 2022b; Mao et al., 2023). Some examples of operator learning frameworks include the Fourier Neural Operator (Li et al., 2021; 2024; White et al., 2023), DeepONet (Lu et al., 2021a; Wang et al., 2021b), In-context operator learning (Yang et al., 2023a), among others, e.g. (O'Leary-Roseberry et al., 2024; Molinaro et al., 2023). However, for solving the inverse problem, neural operators can encounter challenges when the ground truth is out of the distribution of the training dataset.

There are many other methods for PDE inverse problems using deep learning; see (Nganyu Tanyu et al., 2023; Herrmann & Kollmannsberger, 2024; Brunton & Kutz, 2023) for more comprehensive reviews.

## MAIN CONTRIBUTIONS

In this work, we focus on solving PDE inverse problems in the PDE-constrained optimization framework using deep learning methods. The contributions of this paper are as follows:

- We formulate the PDE inverse problem as a bilevel optimization problem, where the upper level problem minimizes the data loss with respect to the PDE parameters, and the lower level problem involves training a neural network to approximate the PDE solution operator locally at given PDE parameters, enabling direct computationi of the descent direction for the upper level optimization problem.

- At the lower level problem, we introduce the "residual-gradient" loss, which is the L2 norm of derivative of the residual with respect to the PDE parameters. We show that this loss term compels the neural network to approximate the PDE solution for a small neighborhood of the PDE parameters, thus a "local operator".

- Extensive experiments over multiple PDE systems demonstrate that our novel formulation is both more accurate and more robust than other existing methods. It exhibits stronger PDE fidelity, robustness to sparse and noisy data, and eliminates the need to balance the residual and the data loss, a common issue in PDE-based soft constraints.

- We solve the bilevel optimization problem using gradient descent simultaneously on both the upper and lower level optimization problems, leading to an effective and fast algorithm. The network architecture is simple and easy to implement.

- We extend our method to infer unknown functions that are also parameterized by neural networks through an auxiliary variable. This bypasses the need to learn a high-dimensional local operator.

Our approach combines elements of PINN, operator learning, and the adjoint method. Our method is closely related to the PINN: both use neural network to represent the solution to the PDE, use automatic differentiation to compute the PDE residual, and aim to solve the PDE and infer the parameters simultaneously. However, in the PINN, the PDE-constraint is enforced as a regularization term (or soft constraint), leading to a trade-off between fitting the data and solving the PDE accurately. Compared with operator learning, which solves the PDE for a wide range of parameters and requires a large amount of synthetic data for training, our method only learns the operator local to the PDE parameters at each step of the optimization process and does not require a synthetic dataset for training. Similar to the adjoint method, we aim to approximate the descent direction for the

PDE parameters with respect to the data loss, but we do not require deriving and solving the adjoint equation.

## 2 METHOD

### 2.1 PDE INVERSE PROBLEM AS BI-LEVEL OPTIMIZATION

In this section, we present a novel method for solving PDE inverse problems in the framework of PDE-constrained optimization problems using deep learning. Let $u : \Omega \to \mathbb{R}$ be a function defined over a domain $\Omega \subset \mathbb{R}^d$ satisfying some boundary conditions, and $\hat{u}$ be the observed data, which might be noisy. Suppose $u$ is governed by a PDE, $F$, which depends on some parameters $\Theta$. Then the following PDE-constrained optimization problem is solved:

$$\min_{\Theta} \quad \|u - \hat{u}\|_2^2 \qquad \text{s.t.} \quad F(D^k u(\mathbf{x}), ..., Du(\mathbf{x}), u(\mathbf{x}), \Theta) = 0 \tag{1}$$

The constraint is a PDE operator that depends on the parameters $\Theta$. For time-dependent problems, we treat time $t$ as a special component of $\mathbf{x}$, and $\Omega$ includes the temporal domain.

Suppose we know the PDE solution operator (hereafter referred to as the "operator"), $u(\mathbf{x}, \Theta)$, which solves the PDE for any $\Theta$, then we can solve the optimization problem easily by minimizing the objective function using a gradient descent algorithm. However, finding the full operator $u(\mathbf{x}, \Theta)$ is challenging and unnecessary. Since we are only interested in the descent direction to update $\Theta$, a local approximation of the solution operator suffices, that is, the operator should approximate the PDE solution for a small neighborhood of a particular value of $\Theta$. For notational simplicity, we define the residual function of the operator as

$$r(\mathbf{x}, \Theta) := F(D^k u(\mathbf{x}, \Theta), ..., Du(\mathbf{x}, \Theta), u(\mathbf{x}, \Theta), \Theta) \tag{2}$$

If $u$ is a local operator at $\Theta$, then $r(\mathbf{x}, \Theta) = 0$ and $\nabla_{\Theta} r(\mathbf{x}, \Theta) = 0$. Our goal is to approximate the operator locally at $\Theta$ using a neural network, and then find the optimal PDE parameters $\Theta$ by minimizing the data loss with respect to $\Theta$ using a gradient descent algorithm.

Suppose the local operator is parameterized by a neural network $u(\mathbf{x}, \Theta; W)$, where $W$ are the weights of the neural network. The objective function (1) leads to the following data loss:

$$\mathcal{L}_{\text{dat}}(\Theta, W) = \frac{1}{|\mathcal{T}_{\text{dat}}|} \sum_{\mathbf{x} \in \mathcal{T}_{\text{dat}}} |u(\mathbf{x}, \Theta; W) - \hat{u}(\mathbf{x})|^2, \tag{3}$$

where $\mathcal{T}_{\text{dat}}$ is the set of collocation points where the data is observed. The residual loss is the L2 norm of the residual function

$$\mathcal{L}_{\text{res}}(W, \Theta) := \frac{1}{|\mathcal{T}_{\text{res}}|} \sum_{\mathbf{x} \in \mathcal{T}_{\text{res}}} |r(\mathbf{x}, \Theta; W)|^2. \tag{4}$$

where $\mathcal{T}_{\text{res}}$ is the set of collocation points where the residual loss is evaluated. We introduce the following loss term, the "residual-gradient loss", which is the derivative of the residual with respect to the PDE parameters $\Theta$:

$$\mathcal{L}_{\text{rgrad}}(\Theta, W) = \frac{1}{|\mathcal{T}_{\text{res}}|} \sum_{\mathbf{x} \in \mathcal{T}_{\text{res}}} |\nabla_{\Theta} r(\mathbf{x}, \Theta)|^2, \tag{5}$$

Intuitively, this loss compels the neural network to approximate the PDE solution for a small neighborhood of $\Theta$: small variation of $\Theta$ should only lead to small variation of the residual. If this is satisfied, then the derivative of the data loss with respect to $\Theta$ will approximate the descent direction, and we can find the optimal $\Theta$ by minimizing the data loss with respect to $\Theta$ using a gradient descent algorithm. We define the "local operator loss" as the sum of the residual loss and the residual-gradient loss with weight $w_{\text{rgrad}}$:

$$\mathcal{L}_{\text{LO}}(\Theta, W) = \mathcal{L}_{\text{res}}(\Theta, W) + w_{\text{rgrad}} \mathcal{L}_{\text{rgrad}}(\Theta, W) \tag{6}$$

Finally, we propose to solve the following bilevel optimization problem:

$$\begin{cases} \Theta^* = \arg\min_{\Theta} \mathcal{L}_{\text{dat}}(\Theta, W^*(\Theta)) \\ W^*(\Theta) = \arg\min_{W} \mathcal{L}_{\text{LO}}(\Theta, W) \end{cases} \tag{7}$$

In the upper level problem, we find the optimal PDE parameters $\Theta$ by minimizing the data loss with respect to $\Theta$. In the lower level problem, we train a network to approximate the local operator $u(\mathbf{x}, \Theta; W)$ by minimizing the local operator loss with respect to the weights of the neural network.

**Pre-train and Fine-tune**  In this work, we assume access to an initial guess of the PDE parameters, $\Theta_0$, alongside their corresponding numerical solution, denoted as $u_0$, e.g. from the finite difference method. The numerical solutions are computed with high accuracy on fine grids, and can be considered as the "exact" solution of the PDE. We can use the numerical solution to pre-train the neural network, and then use the data to fine-tune the neural network to infer the PDE parameters. This has been successfully applied in (Zhang et al., 2024a), and is also similar to curriculum learning, where the neural network learns a "simpler" PDE solution first (Krishnapriyan et al., 2021). We define the pre-training data loss $\mathcal{L}_{u_0}$, which is the MSE between the numerical solution $u_0$ and the local operator at $\Theta_0$:

$$\mathcal{L}_{u_0}(W) = \frac{1}{|\mathcal{T}_{\text{res}}|} \sum_{\mathbf{x} \in \mathcal{T}_{\text{res}}} |u(\mathbf{x}, \Theta_0; W) - u_0(\mathbf{x})|^2, \tag{8}$$

In the pre-training phase, we solve the following minimization problem

$$\min_W \mathcal{L}_{\text{LO}}(\Theta_0, W) + \mathcal{L}_{u_0}(W) \tag{9}$$

The use of $\mathcal{L}_{u_0}$ is not mandatory for training the local operator with fixed $\Theta_0$, though it can speed up the training process.

## 2.2  Inferring an unknown function

We can also extend our method to learn an unknown function $f(\mathbf{x})$ in the PDE, such as a variable diffusion coefficient in the Poisson equation or an initial condition in the heat equation. In these cases, the following PDE constrained optimization problem is solved:

$$\min_f \quad \|u - \hat{u}\|^2 + w_{\text{reg}} \|\nabla f\|^2 \qquad \text{s.t.} \quad F(D^k u(\mathbf{x}), ..., Du(\mathbf{x}), u(\mathbf{x}), f(\mathbf{x})) = 0 \tag{10}$$

where the constraint is a PDE that depends on the unknown function $f$. Given that these problems are ill-posed, regularization of the unknown function is often necessary. A typical choice is the L2-norm of the gradient of the unknown function, which penalizes non-smooth functions. While the selection of an appropriate regularization form is critical and depends on the PDE problem, this paper assumes such choices are predetermined, not an aspect of the method under direct consideration.

Suppose $f$ is parameterized by a neural network $f(\mathbf{x}; V)$ with weights $V$. A straightforward extension from the scalar parameter case is to learn the local operator of the form $u(\mathbf{x}, V)$. However, this would be computationally expensive, as the weights $V$ can be very high dimensional. We propose to introduce an auxiliary variable $z = f(\mathbf{x})$, and find a local operator $u(\mathbf{x}, z)$ such that $u(\mathbf{x}, f(\mathbf{x}))$ solves the PDE locally at $f$. We define the following function $a$, which is the residual function with an auxiliary variable $z$: $a(\mathbf{x}, z) := F(D^k u(\mathbf{x}, z), ..., Du(\mathbf{x}, z), u(\mathbf{x}, z), z)$. If $u$ is a local solution operator at $f$, then we should have: (1) $a(\mathbf{x}, f(\mathbf{x})) = 0$, that the function $u(\mathbf{x}, f(\mathbf{x}))$ have zero residual, and (2) $\nabla_z a(\mathbf{x}, f(\mathbf{x})) = 0$, that small variation of $f$ should lead to small variation of the residual, which has the same interpretation as the parameter inference case (5). These two conditions translates to the corresponding residual loss and residual-gradient loss, similar to (4) and (5). The definitions of the loss functions and the optimization problems are given in Appendix A.

## 2.3  Algorithm

The network architecture involves a simple modification at the input layer (embedding layer) of the typical fully connected neural network: the embedding of the PDE parameters $\Theta$ is randomly initialized and fixed during training, so that the residual-gradient loss can not be made 0 by setting the embedding to 0. See Appendix B for more details.

Solving a bilevel optimization problem is challenging in general (Zhang et al., 2023; Khanduri et al., 2023; Ye et al., 2022; Shen et al., 2023; Shaban et al., 2019; Hong et al., 2022). In our case, the upper level problem (PDE inverse problem) is usually non-convex, and the lower level problem has a challenging loss landscape (Krishnapriyan et al., 2021; Basir & Senocak, 2022a). However, the lower level problem does not need to be solved to optimality at each iteration because the primary goal is to approximate the descent direction for the upper level problem. We propose to apply gradient descent to the upper and lower level optimization problems simultaneously. In Algorithm. 1, we describe our optimization algorithm for inferring scalar parameters in the BiLO

framework. The algorithm for inferring unknown functions is similar. We write the algorithm as simple gradient descent for notational simplicity while in practice we use the ADAM optimizer (Kingma & Ba, 2017).

---

**Algorithm 1** Bi-level Local Operator for inferring scalar PDE parameters

1: **Input:** Collections of collocation points $\mathcal{T}_{\text{res}}$ and $\mathcal{T}_{\text{dat}}$, initial guess of the PDE parameters $\Theta_0$ and the corresponding numerical solution $u_{\text{FDM}}$.
2: **Pre-train:** Solve the following minimization problem

$$\min_W \mathcal{L}_{\text{LO}}(\Theta_0, W) + \mathcal{L}_{\text{u}_0}(W)$$

3: **Fine-Tune:** Simultaneous gradient descent at the upper and lower level (7).

$$\begin{cases} \Theta^{k+1} = \Theta^k - lr_\Theta \nabla_\Theta \mathcal{L}_{\text{dat}}(\Theta^k, W^k) & (11) \\ W^{k+1} = W^k - lr_W \nabla_W \mathcal{L}_{\text{LO}}(\Theta^k, W^k) & (12) \end{cases}$$

---

We can have two different learning rates for the two groups of variables $W$ and $\Theta$, denoted as $lr_W$ and $lr_\Theta$, respectively. We empirically determined $w_{\text{rgrad}} = 0.001$ and $lr_W = lr_\Theta = 0.001$ to be effective across our numerical experiments. It is not imperative for the residual-gradient loss to be minimized excessively; it is sufficient that it approximate the correct descent direction. Under somewhat restrictive assumptions, we are able to obtain a theoretical characterization of the bilevel optimization problem (shown below. See Appendix C for a proof). A more general theoretical understanding of the learning dynamics will be left for future work.

**Proposition:** Assuming (i) the maximum principal holds for the PDE operator; (ii) the parametrized local operator $u(W, \Theta) = g$ on $\partial\Omega$ for all $W$ and $\Theta$; (iii) the lower level problem has a minimizer $W^*(\Theta)$ such that the $u(W^*(\Theta), \Theta)$ is the local operator, then the approximate gradient of the upper level objective at $W^*(\Theta)$ is exact.

## 2.4 DIFFERENCE BETWEEN BILO, PINN, AND NO FOR INVERSE PROBLEMS

**Neural Operator** Neural operators can serve as surrogate models for PDE solution operators, and can be used in algorithms that require solving the forward PDE multiple times, such as Bayesian inference or derivative-free optimization (Kaltenbach et al., 2023; Lu et al., 2022b), or gradient-based optimization algorithms (Zhou et al., 2024; Lu et al., 2022b; Yang et al., 2023b). However, if the objective is to estimate parameters from limited data, the considerable initial cost for data generation and network training might seem excessive. The accuracy of specific PDE solutions depends on the accuracy of the neural operator, and which may decrease if the true PDE parameters fall outside the training data's distribution (de Hoop et al., 2022). Thus, in the context of finding the best estimate of the parameters given the data in a PDE-constrained optimization framework, we mainly compare BiLO with PINNs.

**PINN** Within the PINN framework, the solution of the PDE is represented by a deep neural network $u(\mathbf{x}; W)$, where $W$ denotes all the trainable weights of the neural network (Karniadakis et al., 2021; Raissi et al., 2019; Lu et al., 2021b). Notice that the PDE parameters $\Theta$ are not part of the network input. Therefore the data loss does not depend on the PDE parameters $\Theta$ directly, and we write the data loss as $\mathcal{L}_{\text{dat}}(W)$.

Solving an inverse problem using PINN involves minimizing an unconstrained optimization problem, where the objective function is the weighted sum of the residual loss and the data loss

$$\min_{W,\Theta} \mathcal{L}_{\text{res}}(W, \Theta) + w_{\text{dat}} \mathcal{L}_{\text{dat}}(W) \tag{13}$$

where $w_{\text{dat}}$ is the weight of the data loss. For simplicity of discussion, we assume the weight of the residual loss is always 1. The key feature is that the PDE is enforced as a soft constraint, or as a regularization term for fitting the data. The relationship between the PDE parameter and the data loss is indirect: the descent directions of the PDE parameters are given by $\nabla_\Theta \mathcal{L}_{\text{res}}$, which are independent of the data loss.

**Challenges for PINNs** Solving PDE inverse problems using PINNs can encounter challenges stemming from the soft PDE constraint (13), especially when the data is sparse and noisy, or when the PDE model does not fully explain the data (Zhang et al., 2024a). The soft PDE constraint can result in a trade-off between fitting the data and solving the PDE accurately. In addition, since the PDE parameters are updated in the descent direction of the residual loss, they can be biased toward parameters corresponding to very smooth solutions. It is important to recognize that PINNs can indeed be effective for PDE inverse problems, if the weights are chosen properly or when data is abundant and the noise is independent and identically distributed, as the the minimizer of the data loss still gives a good approximation of the PDE solution.

There are many techniques to improve the performance of PINNs, such as adaptive sampling and weighting of collocation points (Nabian et al., 2021; Wu et al., 2023; Lu et al., 2021b; Anagnostopoulos et al., 2024), new architectures (Jagtap & Karniadakis, 2020; Wang et al., 2024; 2021a; Moseley et al., 2023), new optimization algorithms (Basir & Senocak, 2022b; Krishnapriyan et al., 2021), new loss functions (Wang et al., 2022; Yu et al., 2022; Son et al., 2021), adaptive weighting of loss terms (Maddu et al., 2022; Wang et al., 2021a; McClenny & Braga-Neto, 2022; Wang et al., 2023). However, these techniques do not fundamentally change the soft PDE-constraints in the PINN framework. In our work, we propose a different optimization problem that does not involve a trade-off between the residual loss and the data loss, and our method can be used in conjunction with many of these techniques to improve the performance. Therefore, in the following numerical experiments, we do not use any of these techniques, and we focus on comparing the two different optimization formulations (BiLO and the soft PDE-constraints).

The challenge of balancing trade-offs also motivated BPNHao et al. (2023), which applies a bilevel optimization framework to PDE inverse problems by representing the PDE solution with a neural network, using the residual loss for the lower-level problem, and approximating the upper-level hypergradient with Broyden's method. In contrast, our approach incorporates the PDE parameter as part of the network input, with the lower-level problem focused on approximating the local operator, allowing more direct computation of the upper-level descent direction.

## 3 NUMERICAL EXPERIMENTS

In Section 3.1, we infer two scalar parameters in the Fisher-KPP equation and compare the performance of BiLO, PINN and DeepONet. In Section 3.2, we infer an unknown function in the Poisson equation and compare the performance of BiLO and PINN (results of DeepONet are shown in Appendix E.2). We denote the neural network solution (from BiLO, PINN, or DeepONet) by $u_{\mathrm{NN}}$, and denote the numerical solution with the inferred parameters using the Finite Difference Method (FDM) by $u_{\mathrm{FDM}}$, which is solved to a high accuracy. A large discrepancy between $u_{\mathrm{NN}}$ and $u_{\mathrm{FDM}}$ suggests that the PDE is not solved accurately by the neural network.

We provides the training detail and hyperparameters for the numerical experiments in Section 3.1 and 3.2 in Appendix D. Details of the DeepONet architecture and training are provided in Appendix E. Appendix F provides additional numerical experiments: (1) F.1 Inferring the initial condition of a 1D heat equation; (2) F.2 Inferring the initial condition of a *inviscid* Burger's equation, which is a hyperbolic PDE, and the solution has a shock discontinuity. (3) F.3 Inferring the variable diffusion coefficient of a 2D Poisson problem, where we achieve better or comparable performance as in PINO (Li et al., 2024). Appendix G provides the computational cost for the experiments.

### 3.1 FISHER-KPP EQUATION

In this example, we aim to infer the unknown parameters $D$ and $\rho$ in the following Fisher-KPP equation (Zou et al., 2024), which is a nonlinear reaction-diffusion equation:

$$\begin{cases} u_t(x,t) = 0.01 D u_{xx}(x,t) + \rho u(1-u) \\ u(x,0) = \frac{1}{2}\sin(\pi x)^2 \\ u(0,t) = u(1,t) = 0 \end{cases} \tag{14}$$

The initial guesses of the PDE parameters are $D_0 = 1$ and $\rho_0 = 1$, and the ground truth parameters are $D_{GT} = 2$ and $\rho_{GT} = 2$. This equation has been used to model various biological phenomena, such as the growth of tumors (Swanson et al., 2000; Harpold et al., 2007) or the spreading of

misfolded proteins (Zhang et al., 2024b; Schäfer et al., 2020; 2021). In our tests the data is only provided at the final time $t = 1$, which is more challenging than the case where data is provided at multiple time points. This single-time inference problem has application in patient-specific parameter estimation of tumor growth models using medical images, where only one time point may be available, e.g., in the case of glioblastoma (Balcerak et al., 2024; Zhang et al., 2024a; Ezhov et al., 2023; Scheufele et al., 2021).

**Effect of residual-gradient loss** We plot the trained local operator $u(\mathbf{x}, D_0 + \delta D, \rho_0 + \delta \rho; W)$ at $t = 1$, for $(\delta D, \delta \rho) = (0.5, 0)$ and $(0, 0.1)$, and the corresponding FDM solution in Fig. 1 (a). We can see that even though the network is only trained using the initial parameters, because of the residual-gradient loss, the network can approximate the solution of the PDE for a small neighborhood of the parameters. This suggests that the derivative of the data loss with respect to the parameters should give the correct descent direction.

**Trajectory of the Parameters** We consider the case without noise and show the trajectories of the parameters $D$ and $\rho$ during the fine-tuning process in Fig. 1 (b). Each BiLO trajectory (black line) corresponds to a different random initialization of the neural network, and are obtained by our simultaneous gradient descent. They roughly follow the trajectory that is obtained by solving the lower level problem to a small tolerance before updating the PDE parameters (red dashed line). The contours are the data loss in log scale using the FDM solution for each parameter pair $(D, \rho)$. Note that the contour lines do not represent the actual loss landscape of our optimization problem, since at each step we are not solving the PDE to high accuracy. From the landscape we can also see that single-time inference is challenging, as the gradient with respect to $D$ is much smaller than $\rho$, leading to a narrow valley in the loss landscape along the $D$-direction.

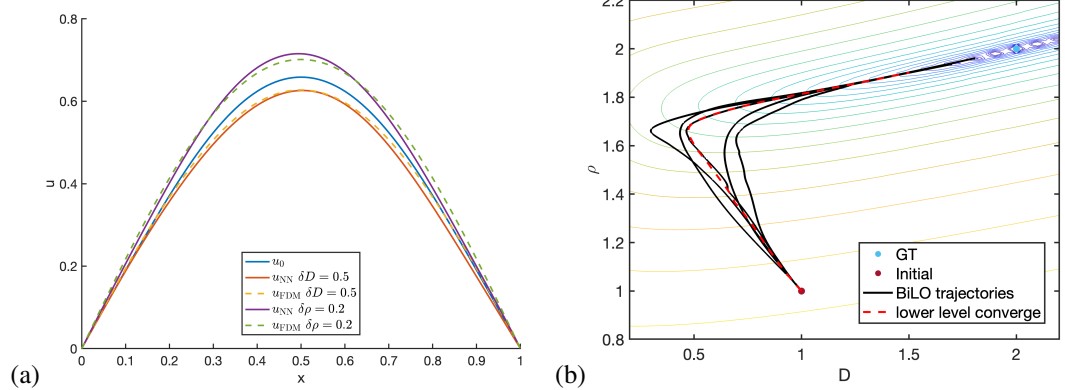

(a)  (b)

Figure 1: (a) Visualization of the local operator $u(\mathbf{x}, D_0 + \delta D, \rho_0 + \delta \rho; W)$ at $t = 1$ for $\delta D = 0.5$ or $\delta \rho = 0.2$, and the corresponding FDM solutions. (b) Trajectory of the parameters $D$ and $\rho$ during fine-tuning roughly follow the path of the steepest descent. The dashed line is the trajectory when the lower level problem is solved to a small tolerance. The contours correspond to the data loss in log scale, computed using the FDM solution.

**Inference with noise** In this experiment, we consider inference under noise $\epsilon \sim N(0, 10^{-4})$. In Fig. 2, we show the results of BiLO and PINNs with different weights $w_{\text{dat}} = 0.01, 0.1, 1$. We can see that for $w_{\text{dat}} = 0.01$ and $0.1$, the PDE is solved relatively accurately, since $u_{\text{NN}}$ and $u_{\text{FDM}}$ overlap. For $w_{\text{dat}} = 1$, the PDE is not solved accurately and the network is over-fitting the data. In addition, PINNs have difficulties in obtaining accurate estimates of $D$ due to the challenging loss landscape. Our new method gives more accurate inferred parameter and PDE solution.

In Table 1, we show the mean and standard deviation (std) of various metrics for BiLO, PINNs with different $w_{\text{dat}}$, and DeepONets with different pretraining datasets. The ground truth solution should have an average data loss of $\mathcal{L}_{\text{dat}} = 10^{-4}$, which is the variance of the noise. We can see that the loss landscape is particularly challenging, leading to relatively large error in $D$ for all methods. For the PINN, we see that $w_{\text{dat}} = 0.01$ leads to under-fitting of the data, as the data loss is larger than the variance of the noise; and $w_{\text{dat}} = 10$ shows clear sign of over-fitting of the data, as the data loss is getting smaller than the variance of the noise. The DeepONets are first pretrained with numerical solutions of the PDE with various $D$ and $\rho$. Then a gradient-based optimization algorithm is used

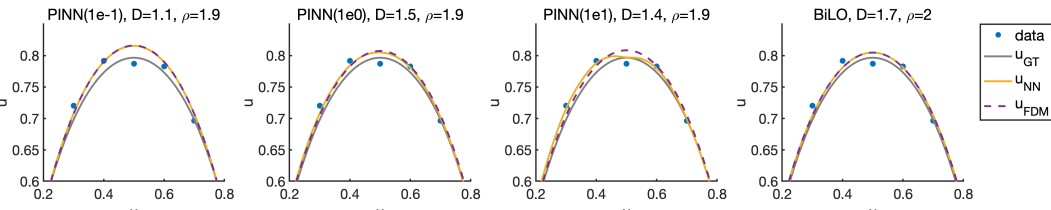

Figure 2: Enlarged view of the network predicted solutions $u_{\mathrm{NN}}$ (BiLO and PINNs with different $w_{\mathrm{dat}}$) and FDM solutions $u_{\mathrm{FDM}}$ at final time, in the region $(x, u) \in [0.2, 0.8] \times [0.6, 0.85]$. BiLO gives more accurate inferred parameters and PDE solution.

to solve the inverse problem. We consider both coarse and dense sampling of the parameters $D$ and $\rho$ that include the ground truth parameters. Additionally, we also consider a dense sampling but the ground truth parameters are out-of-distribution(OOD). Details are provided in Appendix. E.1. We can see that the results from DeepONet are affected by the quality of the pretraining dataset. Overall, BiLO gives more accurate inferred parameters and PDE solution, is robust to the noise, and does not require a large amount of pretraining data.

| method | $|D - D_{GT}|$ | $|\rho - \rho_{GT}|$ | $\|u_{\mathrm{NN}} - u_{\mathrm{FDM}}\|_\infty$ | $\mathcal{L}_{\mathrm{data}}$ |
|---|---|---|---|---|
| BiLO | **0.26±0.10** | **0.06±0.03** | **3.36e-3±1.14e-3** | **1.01e-4±2.77e-5** |
| PINN(1e-1) | 0.85±0.07 | 0.17±0.02 | 9.40e-3±9.15e-4 | 1.43e-4±2.58e-5 |
| PINN(1e0) | 0.40±0.13 | 0.09±0.03 | 4.41e-3±1.44e-3 | 8.68e-5±3.00e-5 |
| PINN(1e1) | 0.44±0.21 | 0.10±0.04 | 4.93e-3±2.10e-3 | 3.29e-5±2.02e-5 |
| DeepONet(Coarse) | 0.95±0.74 | 0.24±0.20 | 7.96e-3±6.36e-3 | 6.26e-5±2.81e-5 |
| DeepONet(Dense) | 0.48±0.40 | 0.13±0.10 | 4.85e-3±3.47e-3 | 6.23e-5±1.95e-5 |
| DeepONet(OOD) | 0.95±0.86 | 0.35±0.38 | 1.62e-2±1.75e-2 | 6.18e-5±1.88e-5 |

Table 1: Comparison of BiLO, PINNs (with various $w_{\mathrm{dat}}$) and DeepONet (with various pretraining dataset) for a Fisher-KPP PDE problem with noise $\epsilon \sim N(0, 10^{-4})$. BiLO gives more accurate inferred parameters and PDE solution.

### 3.2 POISSON EQUATION WITH VARIABLE DIFFUSION COEFFICIENT

In this test, we consider the following Poisson equation on $[0, 1]$ with $u(0) = u(1) = 0$:

$$(D(x)u'(x))' = -\pi^2 \sin(\pi x) \tag{15}$$

and aim to infer the variable diffusion coefficient $D(x)$ such that $D(0) = D(1) = 1$. The ground truth $D(x)$ is a "hat" function $D(x) = 1 + 0.5x$ for $x \in [0, 0.5)$ and $D(x) = 1.5 - 0.5x$ for $x \in [0.5, 1]$. We start with initial guess $D_0(x) = 1$.

**Effect of residual-gradient loss** In Fig. 3, we visualize the local operator $u(x, z; W)$ after pretraining with $D_0(x) = 1$. We consider the variation $\delta D_1(x) = -0.1$, and $\delta D_2(x) = 0.1x$ and evaluate the neural network at $u(x, D_0(x) + \delta D_i(x); W)$ for $i = 1, 2$. The FDM solutions of the PDE corresponding to $D_0(x) + \delta D_i(x)$ are also plotted. We can see that the neural network approximates the solution corresponding to $D_0(x) + \delta D_i(x)$ well.

| method | $\|D - D_{GT}\|_\infty$ | $\|D - D_{GT}\|_2$ | $\|u_{\mathrm{NN}} - u_{\mathrm{FDM}}\|_\infty$ | $\mathcal{L}_{\mathrm{data}}$ |
|---|---|---|---|---|
| BiLO | **5.86e-2±1.99e-2** | **2.01e-2±7.94e-3** | **3.94e-3±1.93e-3** | **1.01e-4±1.80e-5** |
| PINN(1e0) | 9.99e-2±2.88e-3 | 3.97e-2±1.73e-3 | 6.37e-3±1.54e-3 | 1.09e-4±1.85e-5 |
| PINN(1e1) | 8.61e-2±7.50e-3 | 3.25e-2±3.96e-3 | 4.43e-3±1.37e-3 | 1.02e-4±1.87e-5 |
| PINN(1e2) | 7.13e-2±1.59e-2 | 3.11e-2±1.09e-2 | 4.88e-3±1.45e-3 | 9.42e-5±1.55e-5 |
| Adjoint | 7.89e-2±2.27e-2 | 3.12e-2±9.00e-2 | - | 9.14e-4±1.48e-5 |

Table 2: Comparison of BiLO, PINNs (with various $w_{\mathrm{dat}}$) and the adjoint method for inferring a variable diffusion coefficient from noisy data. BiLO is more robust to the noise and gives a more accurate inferred diffusion coefficient and PDE solution.

**Inference With Noise Data** In this experiment, we consider inference under noise $\epsilon \sim N(0, 10^{-4})$ with $w_{\mathrm{reg}} = 10^{-3}$. In Table. 2, we show the mean and standard deviation of various metrics. We

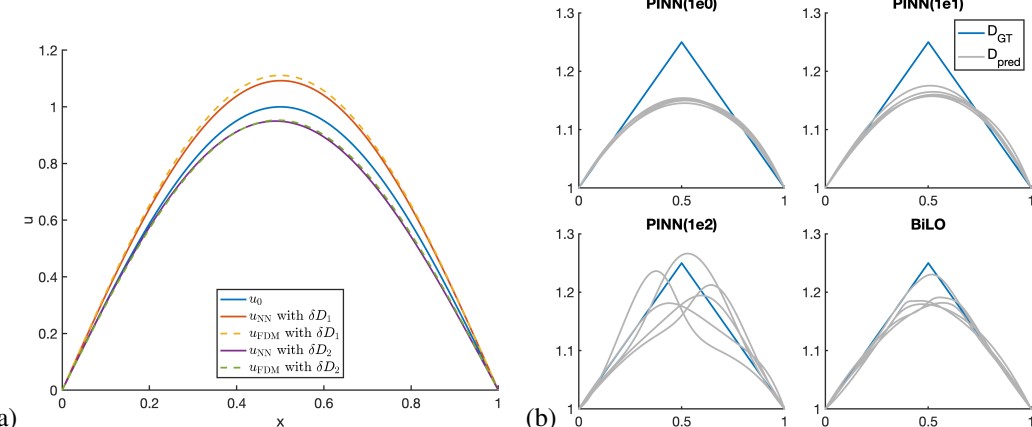

(a)                           (b)

Figure 3: (a) Visualizing the operator $u(x, D(x) + \delta D(x); W)$ after pre-training with $D_0(x) = 1$. (b) Inferred $D(x)$ using noise data with 5 random seeds: BiLO and PINN with various $w_{\text{dat}}$. BiLO gives more accurate inferred $D(x)$.

consider the $L_\infty$ and $L_2$ error of the inferred function $D(x)$ from the ground truth $D_{GT}$, which measure the accuracy of the inferred function; The $L_\infty$ error between $u_{\text{NN}}$ and $u_{\text{FDM}}$ indicates the accuracy of neural network solution; We also show the average data loss $\mathcal{L}_{\text{dat}}$, which ideally should be close to the variance of the noise ($10^{-4}$). A smaller or larger average $\mathcal{L}_{\text{dat}}$ indicates tendencies to over-fit or under-fit the data respectively. For the PINNs, we can see that the optimal $w_{\text{dat}}$ is about 10, as increasing to 100 leads to over-fitting of the data, and decreasing to 0.1 leads to under-fitting of the data. BiLO results in more accurate inferred diffusion coefficient and PDE solution, and is robust to the noise. The inferred $D(x)$ are plotted in Fig. 3 (b). For the PINN, a small $w_{\text{dat}}$ leads to smooth $D(x)$, while a large $w_{\text{dat}}$ leads to an oscillating $D(x)$ due to over-fitting. BiLO gives more accurate inferred $D(x)$ that better approximate the kink of the ground truth $D(x)$. The adjoint methods solved the PDE to high accuracy, but the reconstruction of the diffusion coefficient is not as accurate as BiLO. Appendix E.2.1 describe the adjoint method in more detail, and show the cross validation results with different $w_{\text{reg}}$. In Appendix. E.2.2, we compare BILO with DeepONet, whose performance depends on how we sample the pretraining dataset.

### 3.3 GLIOBLASTOMA (GBM) INVERSE PROBLEM

In this section, we consider a real-world application of BiLO for patient specific parameter estimation of GBM growth models using patient MRI data in 2D. The challenge is that the data are highly noisy, and the model might be misspecified, as the Fisher-KPP PDE may not fully capture the complexities observed in the tumor MRI data. The setup of the problem follows Zhang et al. (2024a); Balcerak et al. (2024); Ezhov et al. (2023); Scheufele et al. (2021).

**Tumor Growth and Imaging Model** Let $\Omega$ be the brain region in 2D based on MRI images. The normalized tumor cell density is $u(\mathbf{x}, t)$.

$$\begin{cases} \frac{\partial u}{\partial t} = D\bar{D}\nabla \cdot (P(\mathbf{x})\nabla u) + \rho\bar{\rho}u(1 - u) & \text{in } \Omega \\ \nabla u \cdot \mathbf{n} = 0 & \text{on } \partial\Omega \end{cases} \tag{16}$$

where $P$ depends on the tissue distribution, and $\bar{D}, \bar{\rho}$ are known patient specific characteristic parameters based on the data. $D$ and $\rho$ are the unknown nondimensionalized parameters that we aim to infer from the data. Let $y^{\text{WT}}$ and $y^{\text{TC}}$ be indicator function of the whole tumor (WT) region and tumor core (TC) region, respectively, which are generated by established segmentation methods. We assume that the segmentations are tumor cell density $u$ at nondimensional $t = 1$ above certain thresholds $u_c^{\text{WT}}$ and $u_c^{\text{TC}}$. The predicted segmentations are given by $y^s_{pred}(\mathbf{x}) = \sigma(20(u(\mathbf{x}, 1) - u_c^s))$, where $\sigma$ is the sigmoid function, for $s \in \{\text{WT}, \text{TC}\}$. We aim to minimize the relative error between the predicted segmentations and segmentation data, under the PDE constraints (16).

$$\min_{D, \rho, u_c^{\text{WT}}, u_c^{\text{TC}}} ||y_{pred}^{TC} - y^{TC}||_2^2/||y^{TC}||_2^2 + ||y_{pred}^{WT} - y^{WT}||_2^2/||y^{WT}||_2^2 \tag{17}$$

**Results** In this scenario, no ground truth is available for the parameters $D$, $\rho$, $u_c^{\mathrm{WT}}$, and $u_c^{\mathrm{TC}}$. We quantify the performance of the inferred parameters by the DICE score between the predicted segmentations and the segmentation data. In table 3, $\mathrm{DICE}_{\mathrm{m}}^{s}$, where $s \in \{\mathrm{WT}, \mathrm{TC}\}$ and $m \in \{\mathrm{FDM}, \mathrm{NN}\}$, denote the DICE scores between the data segmentation and the predicted segmentation based on $u_{\mathrm{FDM}}$ or $u_{\mathrm{NN}}$. $\mathrm{DICE}_{\mathrm{FDM}}$ measure the goodness of the inferred parameters. We also show the relative error of the $u_{\mathrm{NN}}$ and $u_{\mathrm{FDM}}$ at $t = 1$. Fig 4 shows the predicted segmentations using BiLO and PINN with different $w_{\mathrm{dat}}$. For PINN, the DICE score based on $u_{\mathrm{NN}}$ is generally higher than that of $u_{\mathrm{FDM}}$, indicating a tendency to overfit, as seen in the large relative error between $u_{\mathrm{NN}}$ and $u_{\mathrm{FDM}}$. Reducing the data weight $w_{\mathrm{dat}}$ can mitigate this discrepancy. Despite this, the inferred parameters can still have good performance, as shown by the DICE score based on $u_{\mathrm{FDM}}$. In contrast, BiLO provides an accurate PDE solution and well-performing parameters without the need to fine-tune the data weight.

| methods | $\mathrm{DICE}_{\mathrm{NN}}^{\mathrm{WT}}$ | $\mathrm{DICE}_{\mathrm{NN}}^{\mathrm{TC}}$ | $\mathrm{DICE}_{\mathrm{FDM}}^{\mathrm{WT}}$ | $\mathrm{DICE}_{\mathrm{FDM}}^{\mathrm{TC}}$ | $\mathrm{rel.MSE}(\%)$ |
|---|---|---|---|---|---|
| PINN(1e-3) | 0.880 | 0.897 | 0.799 | 0.798 | 9.5 |
| PINN(1e-4) | 0.873 | 0.873 | 0.801 | 0.824 | 6.1 |
| PINN(1e-5) | 0.814 | 0.823 | 0.801 | 0.807 | 0.5 |
| BiLO | 0.809 | 0.807 | 0.809 | 0.800 | 0.3 |

Table 3: Results of the glioblastoma inverse problem. The DICE scores between the data segmentations and the predicted segmentations, based on FDM or neural network. The relative error is computed based on the $u_{\mathrm{NN}}$ and $u_{\mathrm{FDM}}$ at t=1.

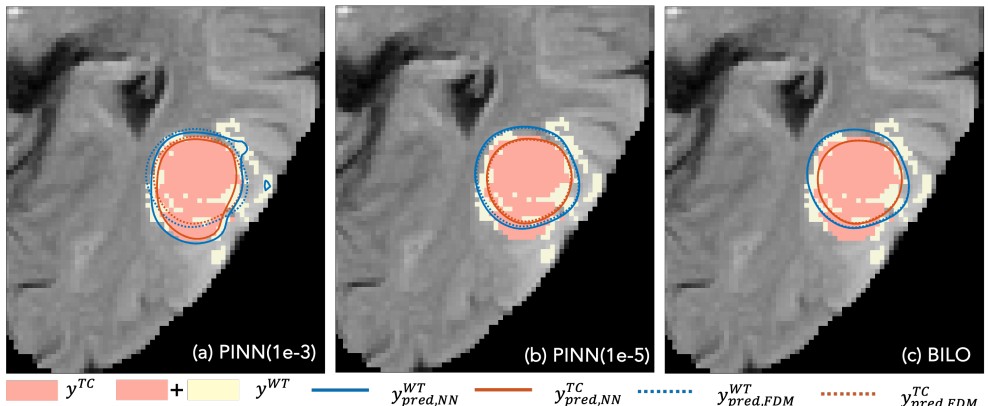

Figure 4: Predicted segmentation using (a) PINN with $w_{\mathrm{dat}}$ = 1e-3 (b) PINN with $w_{\mathrm{dat}}$ = 1e-6 (c) BiLO. The filled regions are the TC and WT region segmentation. The solid and dashed contours are the predicted segmentation using the FDM solution and the neural network solution respectively. BiLO gives almost overlapping contours, indicating a high accuracy of $u_{\mathrm{NN}}$.

## 4 CONCLUSION

In this work, we propose a Bi-level Local Operator (BiLO) learning framework for solving PDE inverse problems: we minimize the data loss with respect to the PDE parameters at the upper level, and learn the local solution operator of the PDE at the lower level. The bi-level optimization problem is solved using simultaneous gradient descent, leading to an efficient algorithm. Empirical results demonstrate more accurate parameter recovery and stronger fidelity to the underlying PDEs under sparse and noisy data, compared with the soft PDE-constraint formulation, which faces the delicate trade-off between adhering to the PDE constraints and accurately fitting the data. As **limitations**: (1) the convergence results are mainly empirical with limited theoretical analysis, (2) the numerical experiments are limited to low dimensional problems, and (3) the architecture of the neural network is simple. Future work includes theoretical analysis of the method, applying the method to more complex and higher dimensional problems, and improving the network architectures.

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

# Appendices

In Appendix A, we define the loss functions for inferring unknown functions in the PDE. In Appendix B, we provide the network architecture of the neural network used in the numerical experiments. In Appendix C, we provide a justification of the simultaneous gradient descent algorithm for the bi-level optimization problem. Appendix D provides the training detail and hyperparameters for the numerical experiments in Section 3.1 and 3.2 in the main text. In Appendix E, we compare BILO with solving PDE inverse problems using a neural operator. Appendix F includes additional numerical experiments

- F.1 Inferring the initial condition of a 1D heat equation.
- F.2 Inferring the initial condition of an inviscid Burger's equation.
- F.3 Inferring the variable diffusion coefficient of a 2D Poisson problem.

Appendix G shows the computational cost of BiLO.

## A  DETAILS FOR INFERRING UNKNOWN FUNCTIONS

As outlined in Section 2.2, suppose $f$ and $u$ are parameterized by neural networks: $f(\mathbf{x}; V)$ and $u(\mathbf{x}; W)$. The data loss is similar to the parameter inference case (3) and depends on both $V$ and $W$. We also need the regularization loss, evaluated on $\mathcal{T}_{\text{reg}}$:

$$\mathcal{L}_{\text{reg}}(V) = \frac{1}{|\mathcal{T}_{\text{reg}}|} \sum_{\mathbf{x} \in \mathcal{T}_{\text{reg}}} |\nabla_{\mathbf{x}} f(\mathbf{x}; V)|^2. \tag{18}$$

We define the residual loss:

$$\mathcal{L}_{\text{res}}(W, V) := \frac{1}{|\mathcal{T}_{\text{res}}|} \sum_{\mathbf{x} \in \mathcal{T}_{\text{res}}} |a(\mathbf{x}, f(\mathbf{x}; V); W)|^2. \tag{19}$$

and the residual-gradient loss:

$$\mathcal{L}_{\text{rgrad}}(W, V) = \frac{1}{|\mathcal{T}_{\text{res}}|} \sum_{\mathbf{x} \in \mathcal{T}_{\text{res}}} |\nabla_z a(\mathbf{x}, f(\mathbf{x}; V); W)|^2 \tag{20}$$

This has the same interpretation as the parameter inference case (5): small variation of $f$ should lead to small variation of the residual. Finally, we solve the following bilevel optimization problem:

$$\begin{cases} V^* = \arg\min_{V} \mathcal{L}_{\text{dat}}(W^*(V), V) + w_{\text{reg}} \mathcal{L}_{\text{reg}}(V) & (21) \\ W^*(V) = \arg\min_{W} \mathcal{L}_{\text{LO}}(W, V) & (22) \end{cases}$$

where $\mathcal{L}_{\text{LO}} = \mathcal{L}_{\text{res}} + w_{\text{rgrad}} \mathcal{L}_{\text{rgrad}}$. At the upper level, we minimize the data loss and the regularization loss with respect to the weights $V$ of the unknown function, and at the lower level, we minimize the local operator loss with respect to the weights $W$ of the local operator. The pre-training stage is similar to the parameter inference case. Given an initial guess of the unknown function $f_0$, and its corresponding numerical solution $u_0$, we can train the network $f_V$ to approximate $f_0$ by minimizing the MSE between $f_V$ and $f_0$, and train the network $u_W$ to be the local operator at $f_0$ by minimizing the local operator loss and the MSE between $u_W$ and $u_0$.

## B  NETWORK ARCHITECTURE

The network architecture involves a simple modification at the input layer (embedding layer) of the typical fully connected neural network. For the scalar parameter case, the input layer maps the inputs $\mathbf{x}$ and the unknown PDE parameters $\Theta$ to a high-dimensional vector $\mathbf{y}$, using an affine transformation followed by a non-linear activation function $\sigma$:

$$\mathbf{y} = \sigma(W\mathbf{x} + R\Theta + \mathbf{b}), \tag{23}$$

where $W$ is the embedding matrix for $\mathbf{x}$, $R$ is the embedding matrix for $\Theta$, and $\mathbf{b}$ is the bias vector. The key is that the embedding matrix $R$ should be non-trainable. Otherwise, $\mathcal{L}_{\text{rgrad}}(W, \Theta)$ can be

made 0 by setting $R$ to be 0. In our work, $R$ will be randomly initialized in the same way as $W$, using uniform distributions in the range of $[-1/\sqrt{d}, 1/\sqrt{d}]$, where $d$ is the number of input units in the layer. The embedding vector $\mathbf{y}$ is then passed through a series of fully connected layers with activation functions. The output of the network is denoted as $\mathcal{N}(\mathbf{x}, \Theta; W)$, where $W$ denotes all the trainable weights of the neural network. In some cases, a final transformation is applied to the output of the neural network $u(\mathbf{x}; W) = \tau(\mathcal{N}(\mathbf{x}, \Theta; W), \mathbf{x})$, to enforce the boundary condition (Dong & Ni, 2021; Lu et al., 2021c; Sukumar & Srivastava, 2022).

## C    SIMULTANEOUS GRADIENT DESCENT

In the main text, we describe the simultaneous gradient descent algorithm for the bi-level optimization problem. In this section, we provide a justification of the algorithm under some assumptions.

We consider the boundary value problem:

$$\begin{cases} \mathbf{L}u = f & \text{in } \Omega \\ u = g & \text{on } \partial\Omega, \end{cases} \tag{24}$$

where $\Omega$ is an connected, open and bounded subset of $\mathbb{R}^d$. $\mathbf{L}$ denoteds a second-order parital differential operator:

$$\mathbf{L}u = \sum_{i,j=1}^{d} a_{ij}\partial_{ij}u + \sum_{i=1}^{d} b_i\partial_i u + cu \tag{25}$$

where the coefficients $a_{ij}$, $b_i$, $c$ are colletively denoted as $\Theta$. We denote $\mathbf{L}_\Theta$ as the derivative of $\mathbf{L}$ with respect to $\Theta$, which is also a differential operator.

We say a function $u(\mathbf{x}, \Theta)$ is a local solution operator of the PDE (25) at $\Theta$ if (1) $\mathbf{L}u = f$ and (2) $L_\Theta u + L\nabla_\Theta u = 0$. That is, the residual at $\Theta$ is zero and the gradient of the residual w.r.t $\Theta$ is zero.

We consider a parameterized local operator $u(\mathbf{x}, \Theta; W)$. For notational simplicity, we omit the dependence of $u$ on $\mathbf{x}$ in the following discussion. We assume that $u(\Theta; W) = g$ on $\partial\Omega$ for all $W$ and $\Theta$.

Our bilevel optimizaiton problem is

$$\min_{\Theta} \int_\Omega (u(\Theta, W^*(\Theta)) - \hat{u})^2 \, d\mathbf{x}$$

$$W^*(\Theta) = \arg\min \int_\Omega (\mathbf{L}u - f)^2 + w_{\text{reg}} (L_\Theta u + L\nabla_\Theta u)^2 \, d\mathbf{x}$$

where $\mathbf{L}u - f$ is the residual of the PDE, and $L_\Theta u + L\nabla_\Theta u$ is the gradient of the residual w.r.t $\Theta$.

In our simultaneous gradient descent, the gradient of the upper level objective with respect to $\Theta$ is given by

$$g_{\text{a}}(W, \Theta) = \int_\Omega (u(W, \Theta) - \hat{u}) (\nabla_\Theta u(W, \Theta)) \, d\mathbf{x} \tag{26}$$

The exact gradient of the upper level objective is

$$g(\Theta) = \int_\Omega (u(W^*(\Theta), \Theta) - \hat{u}) (\nabla_W u(W^*(\Theta), \Theta)\nabla_\Theta W^*(\Theta) + \nabla_\Theta u(W^*(\Theta), \Theta)) \, d\mathbf{x} \tag{27}$$

At $W^*(\Theta)$, the difference between the exact gradient and the approximate gradient, which we denote as $\Delta g$, is given by

$$\begin{aligned} \Delta g(\Theta) :&= g_{\text{a}}(W^*(\Theta), \Theta) - g(\Theta) \\ &= \int_\Omega (u(W^*(\Theta), \Theta) - \hat{u}) (\nabla_W u(W^*(\Theta), \Theta)\nabla_\Theta W^*(\Theta)) \, d\mathbf{x} \end{aligned} \tag{28}$$

Suppose the lower level problem has a minimizer $W^*(\Theta)$ such that the $u(W^*(\Theta), \Theta)$ is the local operator.

$$\mathbf{L}u(W^*(\Theta), \Theta) - f = 0 \tag{29}$$

and

$$\mathbf{L}_\Theta u(W^*(\Theta), \Theta) + \mathbf{L}\nabla_\Theta u(W^*(\Theta), \Theta) = 0 \tag{30}$$

Take the derivative of the Eq. (29) with respect to $\Theta$, we have

$$\mathbf{L}_\Theta u(W^*(\Theta), \Theta) + \mathbf{L}\nabla_\Theta u(W^*(\Theta), \Theta) + \mathbf{L}\nabla_W u(W^*(\Theta), \Theta)\nabla_\Theta W^*(\Theta) = 0 \tag{31}$$

From Eq. (30) and Eq. (31), we have

$$\mathbf{L}\nabla_W u(W^*(\Theta), \Theta)\nabla_\Theta W^*(\Theta) = 0 \tag{32}$$

We denote the function $v := \nabla_W u(W^*(\Theta), \Theta)\nabla_\Theta W^*(\Theta)$. Since $u(W, \Theta) = g$ on $\partial\Omega$ for all $W$ and $\Theta$, we have $v = 0$ on $\partial\Omega$. Therefore, we have $\mathbf{L}v = 0$ in $\Omega$ and $v = 0$ on $\partial\Omega$. If the maximum principal holds for the operator $\mathbf{L}$, for example, when $\mathbf{L}$ uniformly elliptic and $c \geq = 0$, (Evans, 2010) then we have $v = 0$.

By Cauchy-Schwarz inequality, we have

$$||\Delta g||_2 \leq ||u(W^*(\Theta), \Theta) - \hat{u}||_2 ||v||_2 = 0 \tag{33}$$

That is, the approximate gradient at $W^*(\Theta)$ is exact.

We summarize the above discussion in the following proposition:

**Proposition:** Assuming (i) the maximum principal holds for $\mathbf{L}$; (ii) the parametrized local operator $u(W, \Theta) = g$ on $\partial\Omega$ for all $W$ and $\Theta$; (iii) the lower level problem has a minimizer $W^*(\Theta)$ such that the $u(W^*(\Theta), \Theta)$ is the local operator, then the approximate gradient (26) of the upper level objective at $W^*(\Theta)$ is exact.

The assumptions are more restrictive than the numerical experiments. For example, in the Fisher-KPP example, the PDE operator is nonlinear. A more comprehensive and general analysis is left for future work, for example, bounding the error of the approximate gradient by the lower level optimization error (Pedregosa, 2022).

## D  TRAINING DETAILS

For each numerical experiment, we solve the optimization problem 5 times with different random seed. which affect both the initialization of the neural network and the noise in the data (if applicable). Although each realization of the noise may yield a different optimal parameter $\Theta^*$, the average of the optimal parameters across multiple runs should still be close to the ground truth parameter $\Theta_{GT}$. Therefore, we report the mean and standard deviation of the error between the inferred parameters, or functions, and the ground truth quantities.

In all the numerical experiment, we use the tanh activation function and 2 hidden layers, each with 128 neurons, for both PINN and BiLO. The collocation points are evenly spaced as a grid in the domain. For all the optimization problems, we use the Adam optimizer with learning rate 0.001 and run a fixed number of steps.

**Fisher-KPP Equation** Our local operator take the form of $u(x, t, D, \rho; W) = u(x, 0) + \mathcal{N}(x, t, D, \rho; W)x(1 - x)t$ so that the initial condition and the boundary condition are satisfied. Let $X_r$, $X_d$ be the spatial coordinates evenly spaced in $[0, 1]$, and $T_r$ be temporal coordinates evenly spaced in $[0, 1]$. We set $\mathcal{T}_{res} = X_r \times T_r$ and $|X_r| = |T_r| = 51$, that is, the residual collocation points are a uniform grid in space and time. We set $\mathcal{T}_{dat} = X_d \times \{1\}$ and $|X_d| = 11$, that is, the data collocation points form a uniform grid at the final time $t = 1$. Both BiLO and PINN are pretrained with the initial guess for 10,000 steps, and fine-tuned for 50,000 steps.

In Fig. 5, we show the training history of the inferred parameters and the inferred parameters corresponding to Fig 2, and indicate the ground truth with grey dashed line. In Fig. 6 (a), we show the history of the losses of BILO. The data are collected every 20 steps and we applied a moving average with window size 10 to smooth the curves.

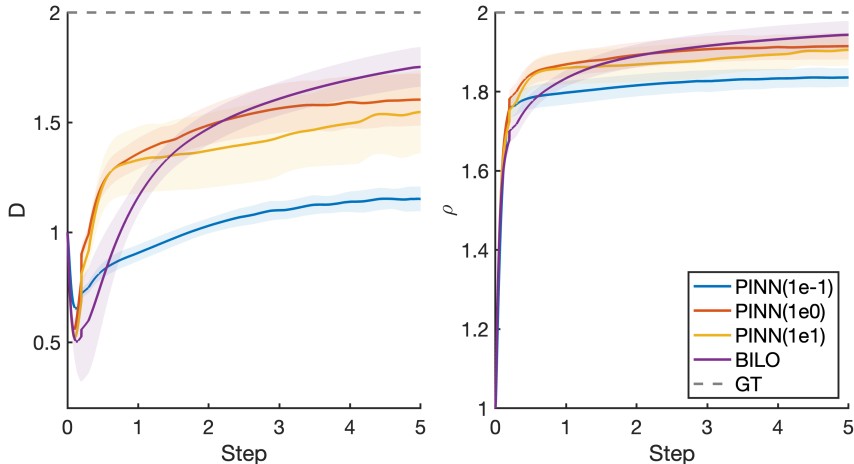

Figure 5: Training history of the inferred parameters corresponding for BiLO and PINN (with various $w_{\text{dat}}$) for the Fisher-KPP equation with noise The solid line is the mean of the inferred parameters across 5 runs, and the shaded region indicates the standard deviation.

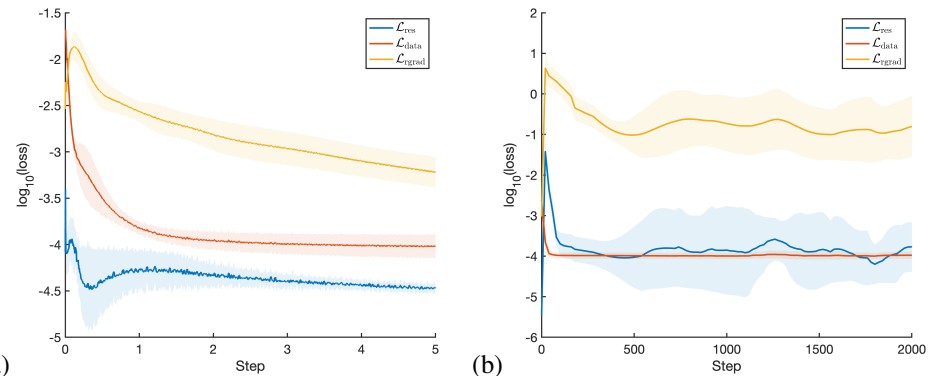

Figure 6: Training history of the unweighted losses ( $\mathcal{L}_{\text{res}}$, $\mathcal{L}_{\text{dat}}$, and $\mathcal{L}_{\text{rgrad}}$ ) during the fine-tuning stage for solving inverse problems using BiLO. The plots includefor (a) the Fisher-KPP equation (Table 1 in Section 3.1) and (b) the Poisson equation with variable diffusion coefficient (Table 2 in Section 3.2, ). Solid lines are the mean of the losses across 5 runs, and the shaded regions indicate the standard deviation.

**Poisson Equation with Variable Diffusion Coefficient**  The local operator takes the form of $u(x, z; W) = \mathcal{N}_1(x, z; W)x(1 - x)$ to enforce the boundary condition, where the fully connected neural network $\mathcal{N}_1$ has 2 hidden layers, each with 128 neurons. The unknown function is parameterized by $D(x; V) = \mathcal{N}_2(x, V)x(1 - x) + 1$, where $\mathcal{N}_2$ has 2 hidden layers, each with 64 neurons. For pre-training, we set $|\mathcal{T}_{\text{res}}| = |\mathcal{T}_{\text{reg}}| = |\mathcal{T}_{\text{dat}}| = 101$, and train 10,000 steps. For fine-tuning, we set $|\mathcal{T}_{\text{res}}| = |\mathcal{T}_{\text{reg}}| = 101$ and $|\mathcal{T}_{\text{dat}}| = 51$, and train 10,000 steps. In Fig. 7, we show the training history of the $\ell_2$ error and the $\ell_\infty$ error of the inferred $D(x)$ for BiLO and PINN (with various $w_{\text{dat}}$). In Fig. 6 (b), we show the history of the losses of BILO.

# E   COMPARISON WITH NEURAL OPERATORS

In this section, we compare the results of BiLO and Neural Operators (NO) for solving the inverse problems. For the NO, we use the DeepONet architecture (Lu et al., 2021a) as an example, which is shown to have comparable performance with FNO (Li et al., 2021; Lu et al., 2022a).

It is difficult to directly compare the performance of NO and PINN/BiLO, since NOs are designed to learn the solution operator of the PDE, while both the PINN and BiLO can be considered as the solver of the PDE, which solve the PDE for one set of parameters. Ususally, NO is trained with a large amount of numerical solutions. In this experiment, for solving the inverse problem, we first

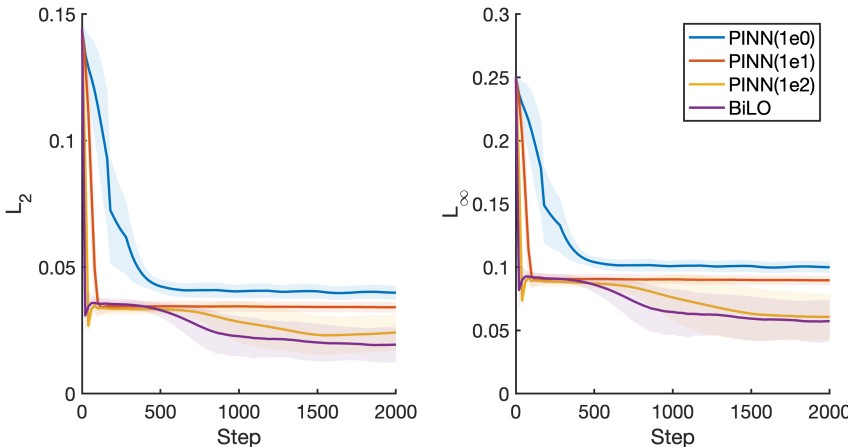

Figure 7: Training history (first 2000 steps) of the $L_2$ error and the $L_\infty$ error of the inferred $D(x)$ for BiLO and PINN (with various $w_{\text{dat}}$) for the Poisson equation with variable diffusion coefficient.

train the NO, and then we use the NO as a surrogate and use gradient-based optimization to infer the parameters of the PDE. We show the the quality of the inferred parameters depends on the quality of the synthetic data used to train the NO. We emphasize that NO can excel in multi-query scenarios, such as sovling the inverse problem in a Bayesian framework, which requires evaluating the solution of the PDE for many different parameters.

### E.1 FISHER-KPP EQUATION

In this experiment, we consider the Fisher-KPP equation with noise, as in Section. 3.1. We consider the following 3 datasets for pretraining the DeepONet. The ground truth parameters are $D_{GT} = 2$ and $\rho_{GT} = 2$, and the initial guess is $D_0 = 1$ and $\rho_0 = 1$. The PDE pararameters are sampled with different range and different resolution. We use the notation $a : h : b$ to denote an array from $a$ to $b$ with step $h$.

- Coarse: $D = 0.8 : 0.05 : 3$, $\rho = 0.8 : 0.05 : 3$.

- Dense: $D = 0.8 : 0.02 : 3$, $\rho = 0.8 : 0.02 : 3$.

- Out-of-distribution (OOD): $D = 0.8 : 0.02 : 1.8$, $\rho = 0.8 : 0.02 : 1.8$.

In the "Coarse" dataset, the parameters are sampled with a larger step size. In the "Dense" dataset, the parameters are sampled with a smaller step size. In the "OOD" dataset, the parameters are sampled with a smaller step size, does not include the ground truth parameters.

We use the following architecture for the DeepONet:

$$G_W(D, \rho, \mathbf{x}) = \sum_{i=1}^{k} b_k(D, \rho) t_k(\mathbf{x})$$

where $b_k(D, \rho)$ is the k-th output of the "branch net", and $t_k(\mathbf{x})$ is the k-th output of the "truck net". Both the trunk net and the truck net are parameterized by fully neural networks with 2 hidden layers, each with 128 neurons, so that the total number of parameters (46179) are comparable to the network used by BILO (42051). The weights of the DeepONet are denoted as $W$. A final transformation on the output $G_W$ is used to enforce the boundary condition. We pre-train multiple DeepONets with 10,000 steps using each datasets.

Given a pretrain dataset with collections of $\{D^j, \rho^j\}$ and their corresponding solutions $u^j$ for $j = 1, \ldots, m$, we first train the DeepONet with the following operator data loss:

$$\min_W \sum_{j=1}^{m} \sum_{\mathbf{x} \in \mathcal{T}_{\text{dat}}} \left| G_W(D^j, \rho^j, \mathbf{x}) - u^j(\mathbf{x}) \right|^2$$

where $\mathcal{T}_{\mathrm{dat}}$ is the same as those used in the BiLO and PINN. For the inverse problem, we fix the weights $W$ and treat the $D$ and $\rho$ as unknown variables. We minimize the data loss:

$$\min_{D,\rho} \frac{1}{|\mathcal{T}_{\mathrm{dat}}|} \sum_{\mathbf{x} \in \mathcal{T}_{\mathrm{dat}}} |G_W(D, \rho, \mathbf{x}) - \hat{u}(\mathbf{x})|^2$$

where $\hat{u}$ is the noisy data.

As shown in Table 1 in the main text, the performance of the inference depends on properties of the pre-training dataset. When the ground truth is out of the distribution of the pre-training dataset, the DeepONet gives poor performance.

### E.2  VARIABLE-DIFFUSION COEFFICIENT POISSON EQUATION

#### E.2.1  IMPLEMENTATION OF THE ADJOINT METHODS

For the numerical example on learning the variable diffusion coefficient of the Poisson Equation, we implement the adjoint method following Vogel (2002). The domain is discretized with uniformly spaced grid points: $x_i = hi$ for $i = 0, \dots, n, n+1$, where $h$ is the spacing of the grid points and $n$ is the number of intervals. We use the finite element discretization with linear basis functions $\phi_i$. Let $\mathbf{u}$ be the nodal value of the solution $u$ at $x_i$ for $i = 1, \dots, n$ and similar for $\mathbf{D}$. We have $\mathbf{u}_0 = \mathbf{u}_{n+1} = 0$ and $\mathbf{D}_0 = \mathbf{D}_{n+1} = 1$. The stiffness matrix $A(\mathbf{D})$ is given by

$$A(\mathbf{D})_{ij} = \frac{1}{2} \begin{cases} \mathbf{D}_{i-1} + 2\mathbf{D}_i + \mathbf{D}_{i+1} & \text{if } i = j \\ -(\mathbf{D}_i + \mathbf{D}_j) & \text{if } |i-j| = 1 \\ 0 & \text{otherwise} \end{cases} \tag{34}$$

The load vector $\mathbf{f}$ is given by $\mathbf{f}_i = f(x_i)$. Suppose the observed data is located at some subset of the grid points of size $m$. Then $\hat{\mathbf{u}} = C\mathbf{u} + \eta$, where $\eta$ is the noise, and $C \in \mathbb{R}^{n \times m}$ is the observation operator. After discretization, the minimization problem is

$$\min_{\mathbf{D}} ||C\mathbf{u} - \hat{\mathbf{u}}||_2^2 + \frac{w_{reg}}{2} \sum_{i=1}^{N} (\mathbf{D}_{i+1} - \mathbf{D}_i)^2$$

$$\text{s.t } A(\mathbf{D})\mathbf{u} = \mathbf{f}$$

The gradient of the loss function with respect to the diffusion coefficient is given by

$$\mathbf{g}_i = \left\langle \frac{\partial A}{\partial D_i} \mathbf{u}, \mathbf{z} \right\rangle + w_{reg} \left( \mathbf{D}_{i+1} - 2\mathbf{D}_i + \mathbf{D}_{i-1} \right)$$

where $\mathbf{z}$ is the solution of the adjoint equation $A^T \mathbf{z} = C^T(C\mathbf{u} - \hat{\mathbf{u}})$. Gradient descent with step size 0.1 is used to update $D$, and is stopped when the norm of the gradient is less than $10^{-6}$.

In table 4, we show the full results of the numerical experiments in Section 3.2 in the main text, with $w_{\mathrm{reg}}$ = 1e-2, 1e-3, 1e-4.

#### E.2.2  COMPARISON WITH DEEPONET

In this experiment, we infer the variable diffusion coefficient $D(x)$ in the Poisson equation using a DeepONet. The pretrain dataset is generated by solving the Poisson equation with 1000 samples of variable diffusion coefficient $D(x)$. $D(x)$ is sampled from a Gaussina Random field on $[0, 1]$, conditioned on $D(0) = D(1) = 1$. The covariance function is the gaussian kernel, with variance 0.05 and different length scale $l = 0.2, 0.3, 0.4$. See Figure 8 for the samples of $D(x)$ and their corresponding solutions. As $l$ increases, the samples of $D(x)$ become smoother.

The DeepONet has the following architecture:

$$G_W(\mathbf{D}, \mathbf{x}) = \sum_{i=1}^{k} b_k(\mathbf{D}) t_k(\mathbf{x})$$

where the vector $\mathbf{D}$ respresent the values of $D(x)$ at the collocation points. A final transformation on the output $G_W$ is used to enforce the boundary condition. In this experiment, both $D$ and $u$ are

| method | $\|D - D_{GT}\|_\infty$ | $\|D - D_{GT}\|_2$ | $\|u_{\mathrm{NN}} - u_{\mathrm{FDM}}\|_\infty$ | $\mathcal{L}_{\mathrm{data}}$ |
|---|---|---|---|---|
| BiLO(1e-2) | 1.08e-1±3.92e-3 | 4.88e-2±2.18e-3 | 2.07e-2±1.28e-3 | 2.57e-4±2.01e-5 |
| BiLO(1e-3) | **5.86e-2±1.99e-2** | **2.01e-2±7.94e-3** | 3.94e-3±1.93e-3 | **1.01e-4±1.80e-5** |
| BiLO(1e-4) | 7.53e-2±1.54e-2 | 2.88e-2±7.91e-3 | 4.30e-3±1.42e-3 | 9.59e-5±1.79e-5 |
| Adjoint(1e-2) | 12.4e-2±3.35e-2 | 5.53e-2±1.45e-2 | - | 8.05e-5±1.29e-5 |
| Adjoint(1e-3) | 7.89e-2±2.27e-2 | 3.12e-2±9.00e-3 | - | 9.14e-5±1.54e-5 |
| Adjoint(1e-4) | 1.09e-1±7.01e-3 | 4.29e-2±3.95e-3 | - | 1.03e-4±1.73e-5 |
| PINN(1e-2/1e0) | 1.62e-1±4.90e-3 | 7.91e-2±2.88e-3 | 3.05e-2±1.46e-3 | 3.85e-4±2.19e-5 |
| PINN(1e-2/1e1) | 1.17e-1±7.67e-3 | 4.83e-2±3.92e-3 | 8.16e-3±1.64e-3 | 1.14e-4±1.97e-5 |
| PINN(1e-2/1e2) | 8.69e-2±1.76e-2 | 3.31e-2±7.99e-3 | 4.18e-3±1.80e-3 | 1.02e-4±1.21e-5 |
| PINN(1e-3/1e0) | 9.99e-2±2.88e-3 | 3.97e-2±1.73e-3 | 6.37e-3±1.54e-3 | 1.09e-4±1.85e-5 |
| PINN(1e-3/1e1) | 8.61e-2±7.50e-3 | 3.25e-2±3.96e-3 | 4.43e-3±1.37e-3 | 1.02e-4±1.87e-5 |
| PINN(1e-3/1e2) | 7.13e-2±1.59e-2 | 3.11e-2±1.09e-2 | 4.88e-3±1.45e-3 | 9.42e-5±1.55e-5 |
| PINN(1e-4/1e0) | 8.69e-2±5.54e-3 | 3.25e-2±2.94e-3 | 4.49e-3±9.97e-4 | 1.04e-4±1.86e-5 |
| PINN(1e-4/1e1) | 7.13e-2±2.08e-2 | 2.64e-2±8.09e-3 | 4.23e-3±1.15e-3 | 9.88e-5±1.72e-5 |
| PINN(1e-4/1e2) | 7.51e-2±2.02e-2 | 3.52e-2±1.45e-2 | 5.19e-3±1.44e-3 | 9.35e-5±1.51e-5 |

Table 4: Comparison of BiLO (with various $w_{\mathrm{reg}}$), Adjoint Method (with various $w_{\mathrm{reg}}$), and PINN (with with various $w_{\mathrm{reg}}/w_{\mathrm{dat}}$)

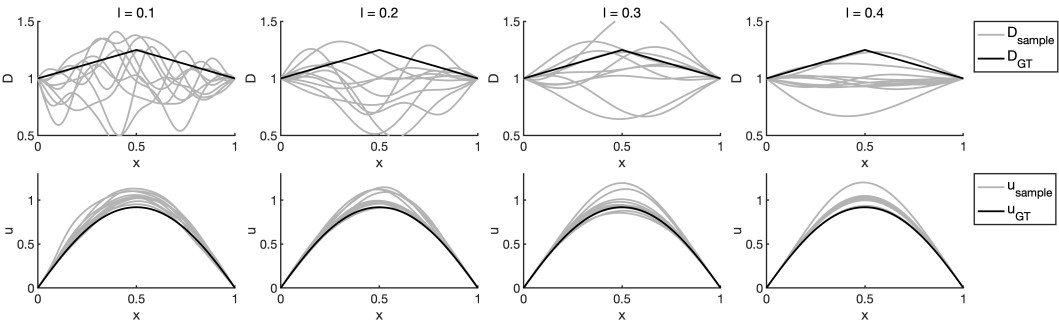

Figure 8: Samples (gray lines) of $D(x)$ with various length scale $l$ and their corresponding solutions. Black line is the ground truth $D$ and $u$

evaluated at 101 points in $[0, 1]$. Let $x_i$ be the collocation points in $[0, 1]$ for $i = 1 = 0, \ldots, N$. Let $\{D^j(x_i), u^j(x_i)\}$ be the samples of $D$ and the corresponding solutions $u$ at $x_i$ for $j = 1, \ldots, m$. We denote $\mathbf{D}^j$ as the vector of $D^j(x_i)$ for $i = 0, \ldots, N$. In the pre-training step, we solve the following minimization problem

$$\min_W \sum_{j=1}^m \sum_{i=1}^N \left| G_W(\mathbf{D}^j, x_i) - u^j(x_i) \right|^2$$

For the inverse problem, we fix the weights $W$ and treat the $\mathbf{D}$ as an unknown variable. We minimize the data loss and a finite difference discretizations of the regularization term $|D(x)|^2$:

$$\min_{\mathbf{D}} \frac{1}{N} \sum_{i=1}^N \left| G_W(\mathbf{D}, x_i) - \hat{u}(x_i) \right|^2 + w_{\mathrm{reg}} \sum_{i=0}^N \left| (\mathbf{D}_{i+1} - \mathbf{D}_i)/h \right|^2$$

where $h$ is the spacing of the collocation points, $\mathbf{D}_0 = \mathbf{D}_N = 1$. Here we work with the vector $\mathbf{D}$ for simplicity. Althernatively, we can represent $D(x)$ as a neural network as in PINN and BiLO experiments.

We perform a grid search on the hyperparameters $l = 0.1, 0.2, 0.3, 0.4$ and $w_{\mathrm{reg}}$=1e-3, 1e-4, 1e-5. In Table 5, we show the 3 combinations of $l$ and $w_{\mathrm{reg}}$ with the best performance in terms of the $L_2$ error of the inferred $D(x)$ and the ground truth. As shown in Table 5, the performance of the inference depends on properties of the pre-training dataset. In practice, it might be difficult to know what does the ground truth unkown function look like. This highlights the importance of the residual loss used in BiLO and PINN, which can help to learn the solution of the PDE without prior knowledge of the ground truth solution.

| method | $\|D - D_{GT}\|_\infty$ | $\|D - D_{GT}\|_2$ | $\|u_{NN} - u_{FDM}\|_\infty$ | $\mathcal{L}_{data}$ |
|---|---|---|---|---|
| BiLO | 5.86e-2±1.99e-2 | **2.01e-2±7.94e-3** | **3.94e-3±1.93e-3** | **1.01e-4±1.80e-5** |
| DeepONet(0.2/1e-5) | **5.55e-2±7.99e-3** | 2.36e-2±2.05e-3 | 6.56e-3±2.36e-3 | 9.45e-5±1.47e-5 |
| DeepONet(0.4/1e-5) | 6.83e-2±2.76e-2 | 2.94e-2±1.14e-2 | 8.65e-3±9.37e-4 | 8.62e-5±1.36e-5 |
| DeepONet(0.4/1e-4) | 8.22e-2±2.08e-2 | 3.16e-2±8.63e-3 | 7.73e-3±1.89e-3 | 1.01e-4±1.83e-5 |

Table 5: Comparison of BiLO and DeepONets ($l$ / $w_{reg}$) pre-trained with datasets with different length scale $l$ and regularization weight $w_{reg}$.

# F  ADDITIONAL NUMERICAL EXPERIMENTS

## F.1  INFER THE INITIAL CONDITION OF A HEAT EQUATION

In this example, we aim to infer the initial condition of a 1D heat equation from the final state. Consider the heat equation

$$\begin{cases} u_t(x,t) = Du_{xx}(x,t) \\ u(x,0) = f(x) \\ u(0,t) = u(1,t) = 0 \end{cases} \tag{35}$$

on $x \in [0,1]$ and $t \in [0,1]$, with fixed diffusion coefficient $D = 0.01$, and unknown initial condition $f(x)$, where $f(0) = f(1) = 0$. Our goal is to infer the initial condition $f(x)$ from observation of the final state $u(x,1)$. We set the ground truth initial condition $f_{GT}$ to be the hat function

$$f_{GT}(x) = \begin{cases} 2x, & \text{if } x \in [0,0.5) \\ 2 - 2x, & \text{if } x \in [0.5,1] \end{cases} \tag{36}$$

We set the initial guess $f_0(x) = \sin(\pi x)$. We can represent the unknown function $f(x;V) = s(\mathcal{N}(x;V))x(1-x)$, where $N_f$ is a fully connected neural network with 2 hidden layers and width 64, and $s$ is the softplus activation function (i.e., $s(x) = \log(1 + \exp(x))$). The transformation ensures that the initial condition satisfies the boundary condition and is non-negative. For BiLO, the neural network is represented as $u(x,t,z) = N_u(x,t,z;W)x(1-x)t + z$, where $N_u$ is a fully connected neural network with 2 hidden layers and width 128. For the PINN, we have $u(x,t;W,V) = N_u(x,t;W)x(1-x)t + f(x;V)$. These transformations ensure that the networks satisfy the boundary and initial condition.

Let $X_r$, $X_d$ be spatial coordinates evenly spaced in $[0,1]$ and $T_r$ be temporal coordinates evenly spaced in $[0,1]$ (both including the boundary). We set $\mathcal{T}_{res} = X_r \times T_r$ and $|X_r| = |T_r| = 51$. That is, the residual collocation points is a uniform grid in space and time. We set $\mathcal{T}_{dat} = X_d \times \{1\}$ and $|X_d| = 11$. That is, the data collocation points is a uniform grid in space at the final time $t = 1$. We set the collocation point for the regularization loss of the unknown function $\mathcal{T}_{reg}$ to be 101 evenly spaced points in the spatial domain.

To evaluate the performance of the inferred initial condition $f$, we use the $L_2$ norm and the $L_\infty$ norm of the difference between the inferred initial condition and the ground truth initial condition, which are evaluated at 1001 evenly spaced points in the spatial domain.

### WITHOUT NOISE

First we consider the case where the data is provided at $t = 1$ without noise. In this case, we also do not use regularization term for the initial condition. In Fig. 9, and Table 6, we show the results of PINNs various weights $w_{dat}$= 0.1, 10, 1000, and BiLO. We can see that BiLO achieved the best $e_2$ and $e_\infty$, demonstrating the effectiveness in recovering the non-smooth initial condition. With very large data loss, the error of the PINN increases. This is because data is only provided at the final time, we need to solve the PDE accurately to infer the initial condition.

### WITH NOISE

In this experiment, we consider the case with noise $\epsilon \sim N(0, 0.001)$. Due to the ill-posedness of the inverse problem, we need to regularize the problem by the 2-norm of the derivative of the unknown function with $w_{reg} = 1e - 2$. In Fig. 10 and Table 7, we show examples of the inferred initial condition and the PDE solution for the PINN formulation with various $w_{dat}$. In Table 7, for

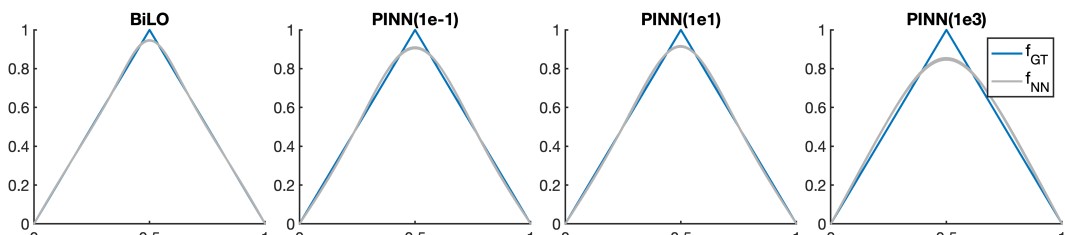

Figure 9: Predicted initial conditions of the heat equation (without noise) from 5 random seeds.

| method | $\|f_{\mathrm{NN}} - f_{\mathrm{GT}}\|_\infty$ | $\|f_{\mathrm{NN}} - f_{\mathrm{GT}}\|_2$ | $\|u_{\mathrm{NN}} - u_{\mathrm{FDM}}\|_\infty$ | $\mathcal{L}_{\mathrm{data}}$ |
|---|---|---|---|---|
| BiLO | **5.43e-2±1.00e-3** | **1.01e-4±7.46e-6** | **4.64e-4±2.57e-4** | **1.52e-9±5.89e-10** |
| PINN(1e-1) | 9.24e-2±2.21e-3 | 5.43e-4±2.81e-5 | 1.29e-3±1.53e-3 | 4.05e-6±4.57e-6 |
| PINN(1e1) | 8.69e-2±2.39e-3 | 4.31e-4±5.06e-5 | 2.44e-3±9.86e-4 | 1.62e-6±1.82e-6 |
| PINN(1e3) | 1.49e-1±4.19e-3 | 1.92e-3±1.34e-4 | 2.54e-2±2.92e-3 | 3.83e-8±5.36e-8 |

Table 6: Comparison of BiLO and PINNs (with various $w_{\mathrm{dat}}$) for inferring the unknown inititial condition (without noise), showing mean (std).

the PINN, we can see that as $w_{\mathrm{dat}}$ increase from 0.1 to 10, it seems that the reconstruction error decreases. However, the $\mathcal{L}_{\mathrm{dat}}$ is becoming smaller than the variance of the noise, indicating that the PINN is overfitting the data. This can also be observed from the Fig 10, for $w_{\mathrm{dat}} = 1e3$, we see larger discrepancy between $u_{\mathrm{PINN}}$ and $u_{\mathrm{FDM}}$.

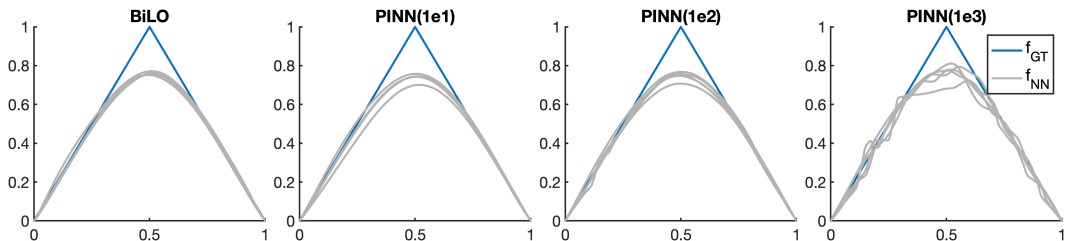

Figure 10: Predicted initial condition $f(x)$ by BiLO and PINNs with various $w_{\mathrm{dat}}$.

| method | $\|f_{\mathrm{NN}} - f_{\mathrm{GT}}\|_\infty$ | $\|f_{\mathrm{NN}} - f_{\mathrm{GT}}\|_2$ | $\|u_{\mathrm{NN}} - u_{\mathrm{FDM}}\|_\infty$ | $\mathcal{L}_{\mathrm{data}}$ |
|---|---|---|---|---|
| BiLO | **2.41e-1±7.62e-3** | **6.11e-3±5.36e-4** | **1.46e-3±7.50e-4** | 4.08e-3±2.53e-4 |
| PINN(1e1) | 2.62e-1±2.22e-2 | 8.13e-3±2.79e-3 | 1.26e-1±3.27e-2 | **5.21e-4±1.55e-4** |
| PINN(1e2) | 2.53e-1±2.34e-2 | 7.11e-3±2.06e-3 | 1.38e-1±3.17e-2 | 2.61e-4±1.56e-4 |
| PINN(1e3) | 2.42e-1±4.65e-2 | 6.56e-3±2.70e-3 | 1.36e-1±3.33e-2 | 2.05e-4±1.65e-4 |

Table 7: Comparison of the BiLO and PINN (with various $w_{\mathrm{dat}}$) for a heat equation with unknown inititial condition (noise $\epsilon \sim N(0, 0.001)$), showing mean (std).

### F.2 INFERRING INITIAL CONDITION OF INVISCID BURGER'S EQUATION

We consdier an inverse problem governed by an inviscid Burger's equation on the domain $x \in [0, 1]$ and $t \in [0, 1]$.

$$\begin{cases} u_t + auu_x = 0 \\ u(x, 0) = f(x) \\ u(0, t) = u(1, t) = 0 \end{cases} \tag{37}$$

where a = 0.2. We aim to infer the initial condition $f$ from the observational data at $t = 1$. The numerical solutions are computed by using the Godunov scheme. The invscid Burger's equation is a hyperbolic PDE, and the solution can develop shocks and rarefraction waves.

We present two examples, as shown in Fig. 11 and Fig. 12. In both examples, the initial guess is $f(x) = 1 - \cos(2\pi x)$. which leads to a mostly smooth solution in the time interval $[0, 1]$. In example

1 (fig f:burger1), the ground truth solution corresponds to the initial condition $f(x) = \sin(2\pi x)$. In example 2 (fig f:burger2), the ground truth solution corresponds to the initial condition $f(x) = -\cos(2\pi x)$ for $x \in [\pi/4, 3\pi/4]$ and $f(x) = 0$ otherwise. Notice that both solutions develop shocks and rarefraction waves, and thus the solution is non-smooth.

In Fig. 11 and Fig. 12, we show the initial guess in the first column, the ground truth in the second column, and the inferrence results by BiLO in the third column. The first row shows the initial condition $f(x)$, the second rows shows the solution $u(x,t)$ on the domain $x \in [0,1]$ and $t \in [0,1]$, and the thrid row shows the solution $u(x,1)$. Notice that for inference, only solution at $t = 1$ of the ground truth is provided. We can see that the BiLO can accurately infer the initial condition of the Burger's equation, even when the solution is non-smooth.

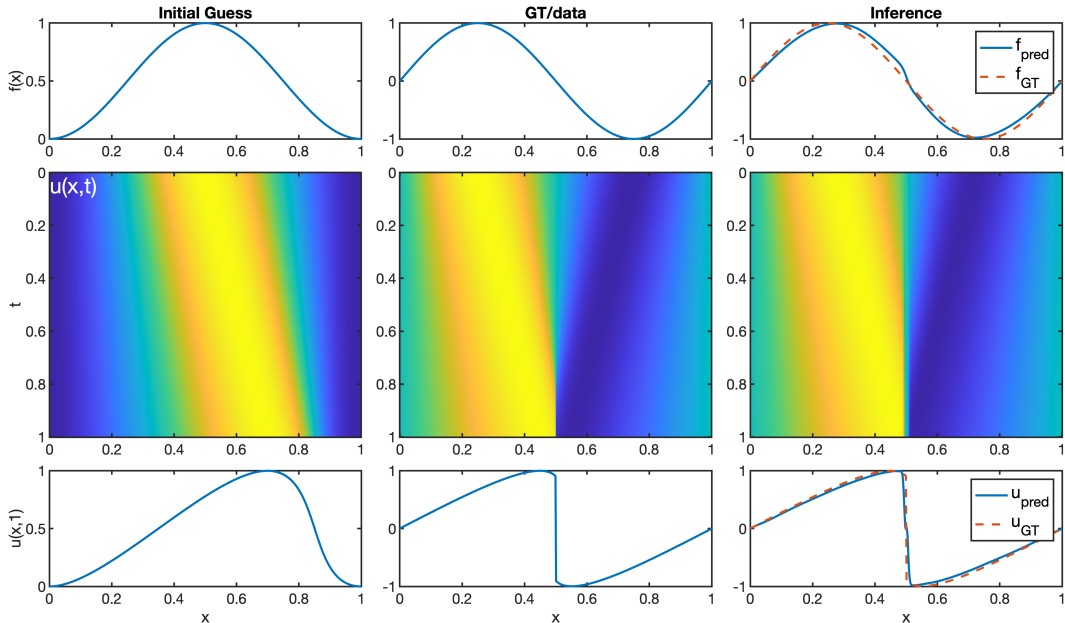

Figure 11: Example 1 of inferring the initial condition of the Burger's equation. The initial guess is used to pre-train the network. The solution at $t = 1$ of the GT is the data for inference. First column: initial guess, second column: ground truth, third column: inferred initial condition. Fisrt row: initial condition, second row: solution $u(x,t)$, third row: solution $u(x,1)$.

### F.3    2D POISSON EQUATION WITH VARIABLE DIFFUSION COEFFICIENT

The setup of this experiment is similar to the steady state Darcy flow inverse problem in (Li et al., 2024). We consider the following 2D Poisson equation with variable diffusion coefficient in the unit square domain $\Omega = [0,1] \times [0,1]$ with Dirichlet boundary condition:

$$\begin{cases} -\nabla \cdot (A(\mathbf{x})\nabla u(\mathbf{x})) = f(\mathbf{x}) & \text{in } \Omega \\ u(\mathbf{x}) = 0, & \text{on } \partial\Omega \end{cases} \tag{38}$$

Our goal is to infer the variable diffusion coefficient $A(\mathbf{x})$ from the solution $u(\mathbf{x})$.

Let $\phi(\mathbf{x})$ be samples of a Gaussian random field (GRF) with mean 0 and squared exponential (Gaussian) covariance structure $C(\mathbf{x},\mathbf{y}) = \sigma \exp(-||\mathbf{x}-\mathbf{y}||^2/\lambda^2)$, where the marginal standard deviation $\sigma = \sqrt{10}$ and the correlation length $l = 0.01$ (Constantine, 2024). This GRF is different from (Li et al., 2024). We generate the initial guess $A_0(\mathbf{x}) = \text{sigmoid}(\phi_0(\mathbf{x})) \times 9 + 3$, where $\phi_0(\mathbf{x})$ is a sample of the GRF. We consider the ground truth diffusion coefficient to be a piece-wise constant function: $A_{\text{GT}}(\mathbf{x}) = 12$ if $\phi_{\text{GT}}(\mathbf{x}) > 0$ and $A_{\text{GT}}(\mathbf{x}) = 3$ otherwise, where $\phi_{\text{GT}}$ is another sample of the GRF. The corresponding solution of $A_0$ and $A_{\text{GT}}$ are denoted as $u_0$ and $u_{\text{GT}}$.

We pretrain the BiLO with $A_0(\mathbf{x})$ and it's corresponding solution $u_0(\mathbf{x})$ for 10,000 steps. And we fine-tune the BiLO for 5,000 steps using $u_{\text{GT}}(\mathbf{x})$ to infer $A_{\text{GT}}$. Following (Li et al., 2024), we use the total variation regularization $|\nabla A|$ with weight $w_{\text{reg}} = 1e - 9$.

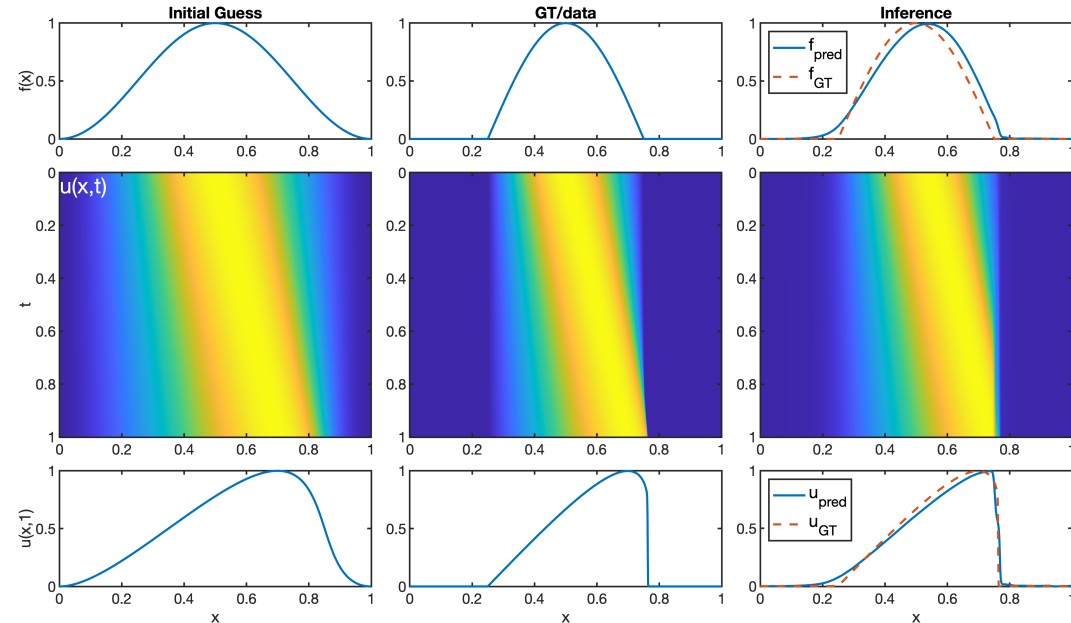

Figure 12: Example 2 of inferring the initial condition of the Burger's equation. The initial guess is used to pre-train the network. The solution at $t = 1$ of the GT is the data for inference. First column: initial guess, second column: ground truth, third column: inferred initial condition. Fisrt row: initial condition, second row: solution $u(x, t)$, third row: solution $u(x, 1)$.

The unknown function is represendted $A(\mathbf{x}; V) = s(\mathcal{N}(\mathbf{x}; V)) \times 9 + 3$, where $N_f$ is a fully connected neural network with 2 hidden layers and width 64, and $s$ is the sigmoid activation function (i.e., $s(u) = 1/(1 + \exp(-u))$). The transformation is a smoothed approximation of the piece-wise constant function. For BiLO, the neural network is represented as $u(\mathbf{x}, z) = N_u(\mathbf{x}, z; W)\mathbf{x}_1(1 - \mathbf{x}_1)\mathbf{x}_2(1 - \mathbf{x}_2)$, where $N_u$ is a fully connected neural network with 2 hidden layers and width 128, and $z$ is our auxiliary variable such that $z = A(\mathbf{x}; V)$.

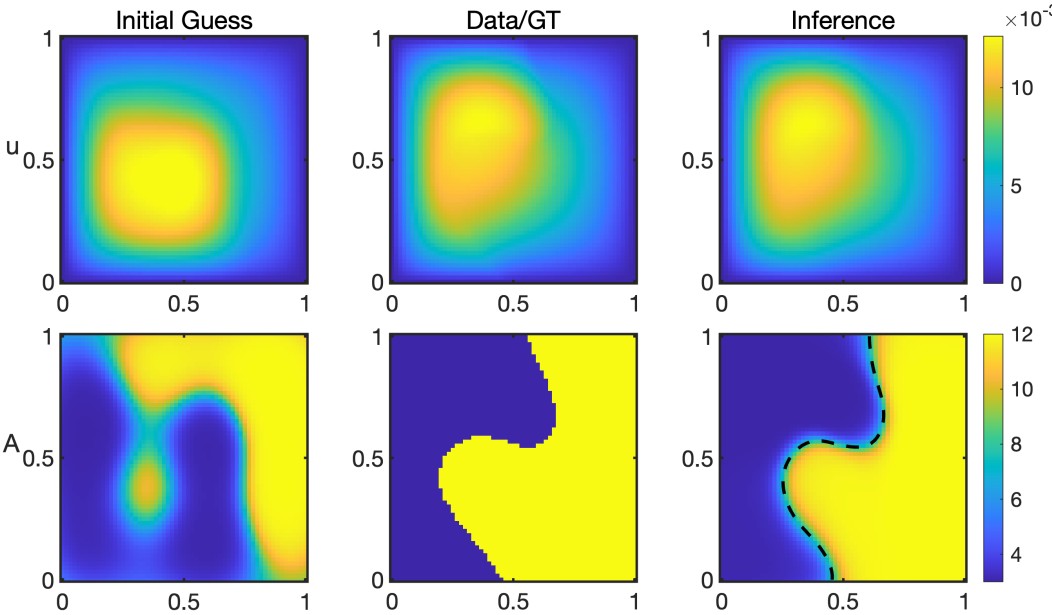

Figure 13: Example 1 of inferring the variable diffusion coefficient. The relative l2 error of $u_{\mathrm{NN}}$ against $u_{\mathrm{GT}}$ is 1.3%. The thresholded (at the dashed line) inferred diffusion coefficient has classification accuracy of 98%

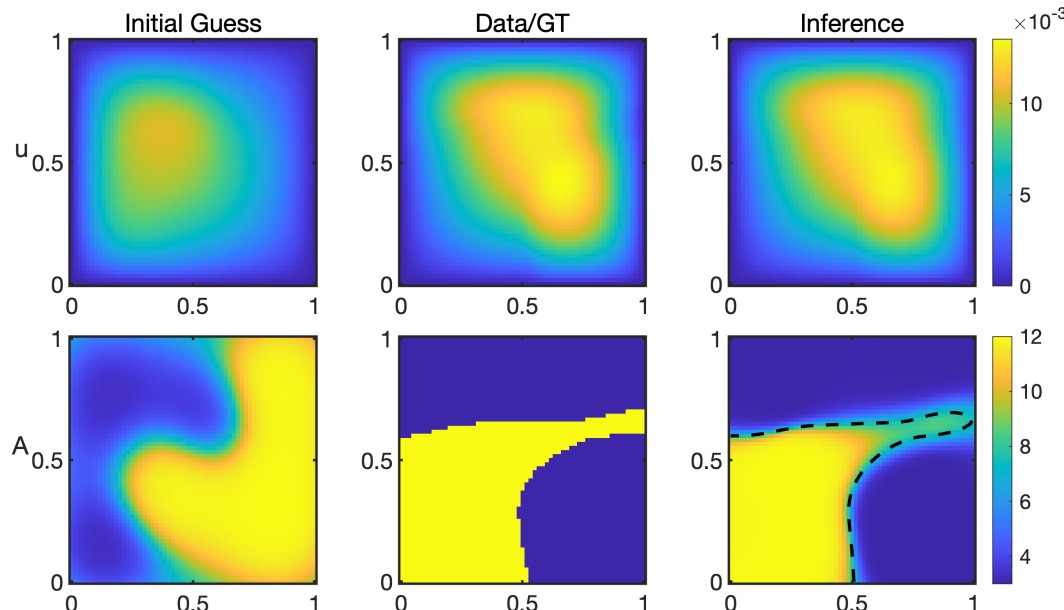

Figure 14: Example 2 of inferring the variable diffusion coefficient. The relative l2 error of $u_{\mathrm{NN}}$ against $u_{\mathrm{GT}}$ is 1.7%. The thresholded (at the dashed line) inferred diffusion coefficient has classification accuracy of 96%

In Figure 13 and Figure 14, we show two examples of the results, each with different initial guess $A_0$ and ground truth $A_{\mathrm{GT}}$. In example 1 (see Figure 13), the relative error of the inferred diffusion coefficient is 1.3%. If we threshold the inferred diffusion coefficient at 7.5 (the mid-point of 3 and 12), the classification accuracy is 98%. In example 2 (see Figure 14), the relative error of the inferred diffusion coefficient is 1.7%. If we threshold the inferred diffusion coefficient, the classification accuracy is 96%. Our performance is comparable to the results (2.29% relative l2 error on $u$ and 97.10% classification accuracy) from the Physics-informed Neural Operator (PINO) in (Li et al., 2024), which require pretraining a FNO with synthetic dataset, and instance-wise fine-tuning with physics-informed loss. For our method, we only need to pretrain the BiLO with a single initial guess. In addition, as shown in the figures, the intial guess can be very different from the ground truth.

## G    COMPUTATIONAL COST

Compared with PINN, BiLO involve computing a higher order derivative term in the residual-gradient loss. This increases the memory cost and computation time per step. However, as shown in Fig. 5, BiLO might require fewer iterations to achieve certain accuracy of the parameters.

In Table. 8, we show the seconds-per-step and the maximum memory allocation of 1 run of BiLO and PINN for the various problems. The seconds per step is computed by total training time divided by the number of steps. The maximum memory allocation is the peak memory usage during the training. For for all the experiments, we use Quadro RTX 8000 GPU. We note that the measured seconds-per-step is not subject to rigorous control as the GPU is shared with other users and many runs are performed simultaneously. Detailed study of the computational efficiency of BiLO will be left for future work.

It is not straightforward to comparing the computational cost with Neural operators. Neural operators can be very fast in the inference stage (solving inverse problem). However, they have significant overhead, which involve preparing the training data, that is, solve the PDE numerically for a large collection of parameters, and pre-train the neural network. The overall cost might be favorable in the many-query settings. However, if we aim to solve the inverse problem once, the total computational cost might not be favorable.

| Problem | Metric | BiLO | PINN | BiLO/PINN |
|---------|--------|------|------|-----------|
| Fisher-KPP | sec-per-step | 0.074 | 0.045 | 1.65 |
| | max-mem-alloc | 200.3 | 65.1 | 3.07 |
| 1D Poisson | sec-per-step | 0.064 | 0.037 | 1.72 |
| | max-mem-alloc | 23 | 20 | 1.15 |
| Heat | sec-per-step | 0.070 | 0.045 | 1.56 |
| | max-mem-alloc | 210 | 109 | 1.92 |

Table 8: Example of computational cost of BiLO and PINN and their ratio for various problems.

## H    COMPARISON WITH BPN

In PINN and BPN (Hao et al., 2023), the PDE solution is represented by a neural network $u(x; W)$. Notice that $\Theta$ is not an input to the neural network. In both PINN and BPN, the data loss is given by

$$L_{data}(W) = \frac{1}{N} \sum_i (u(x_i; W) - \hat{u}_i)^2$$

and enforce the PDE constraints by minimizing the residual loss.

$$L_{res}(W, \Theta) = \frac{1}{N} \sum_i F(D^k u(x_i; W), ..., u(x_i; W), \Theta)^2.$$

Motivated by the same concern about the trade-off between the data loss and the PDE loss in a penalty-like formualtion in PINN. In BPN, the residual loss is separate from the data loss, leading to the bilevel optimization problem

$$\min_{\Theta} L_{data}(W^*(\Theta))$$
$$\text{s.t.} \quad W^*(\Theta) = \arg \min_W L_{res}(W, \Theta).$$

Notice that $L_{data}$ depends on $\Theta$ in directly through the minimizer of lower level problem. The gradient of the data loss with respect to the PDE parameters is given by the chain rule

$$\frac{\mathrm{d}L_{data}}{\mathrm{d}\Theta} = \frac{\mathrm{d}L_{data}(W^*(\Theta))}{\mathrm{d}W} \frac{\mathrm{d}W^*(\Theta)}{\mathrm{d}\Theta},$$

where the hypergradient is given by

$$\frac{\mathrm{d}W^*(\Theta)}{\mathrm{d}\Theta} = -\left[ \frac{\partial^2 L_{res}}{\partial W \partial W^T} \right]^{-1} \cdot \frac{\partial^2 L_{res}}{\partial W \partial \Theta^T}.$$

Broyden's method (Broyden, 1965) is used to compute the hyper-gradient, which is based on the low-rank approximation of the inverse Hessian. In BPN, the bilevel optimization problem is solved iteratively. At each step, gradient descent is performed at the lower level for a fixed number of iterations, $N_f$. Following this, the hypergradient is computed using Broyden's method, which requires $r$ iterations to approximate the inverse vector-Hessian product. This hypergradient is then used to perform a single step of gradient descent at the upper level.

The BiLO approach differs significantly. Instead of representing the PDE solution, BiLO represents the local PDE operator, leading to a different lower level problem that includes the residual-gradient loss. In addition, as the local PDE operator includes $\Theta$ as an input, the data loss depends on the PDE parameters directly:

$$L_{data}(W, \Theta) = \frac{1}{N} \sum_i (u(x_i, \Theta; W) - \hat{u}_i)^2.$$

This enables direct computation of gradients for $Ldata$ with respect to $\Theta$, eliminating the need for specialized algorithms to approximate th hypergradient. The residual-gradient loss also ensures that this direction is a descent direction. This formulation also allows us to perform simultaneous gradient descent at the upper and lower levels, which is more efficient than the iterative approach in BPN. Our method is specialized for PDE-constrained optimization, leveraging the structure of the PDE constraint for efficiency (see the proposition in Appendix. C). In contrast, BPN adopts a more

general bilevel optimization framework, which, while broadly applicable, does not fully exploit the unique characteristics of PDE problems.

To compare BiLO with BPN, we adopted the problem (39) and the setup from Hao et al. (2023), using the same residual points (64), neural network architecture (4 hidden layers with 50 units), upper-level optimizer (Adam with learning rate 0.05), lower-level optimizer (Adam with learning rate 0.001), and initial guess ($\theta_0 = 0, \theta_1 = 1$). Both methods included 1000 pretraining steps to approximate the PDE solution at initial parameters. In BPN, 64 lower iterations are performed for each upper iteration, with 32 Broyden iterations to compute the hypergradient. By contrast, BiLO performs simultaneous gradient descent at the upper and lower levels, where each iteration updates both levels concurrently.

$$
\min_{\theta_0, \theta_1} J = \int_0^1 \left( y - x^2 \right)^2 dx
$$
$$
\text{s.t.} \quad \frac{d^2 y}{dx^2} = 2, \quad y(0) = \theta_0, \quad y(1) = \theta_1
\tag{39}
$$

Figure 15 presents the loss and the error of the PDE parameters for both methods versus the number of lower-level iterations. BiLO achieves a parameter error below 0.01 in fewer than 80 iterations and just 6.4 seconds, while BPN requires 27 upper iterations (1728 lower iterations) and 231 seconds to reach the same accuracy. While this highlights BiLO's efficiency, we note that both methods may benefit from further hyperparameter tuning, and the comparison is made under the settings reported in (Hao et al., 2023).

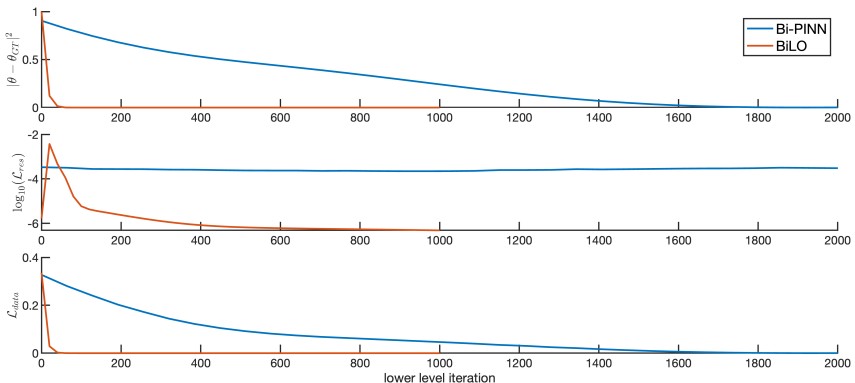

Figure 15: Comparison of BPN and BiLO methods. x-axis is the number of lower level optimization steps. Top: Parameter error $\|\theta - \theta_{GT}\|^2$ versus iterations. Middle: PDE loss $\log_{10}(\mathcal{L}_{res})$. Bottom: Data loss $\mathcal{L}_{data}$.

## I    EFFECT OF NOISE

In this section, we examine the effect of noise in the data on the performance of BiLO and PINN for the Fisher-KPP problem. The residual is evaluated on a $51 \times 51$ grid, while the data is evaluated on an $11 \times 11$ grid in the spatial-temporal domain. Unlike the example in the main text, where data is provided only at the final time, this setup uses observations at all time points, making the problem slightly easier and allowing for fewer fine-tuning steps to achieve convergence. Both BiLO and PINN are pretrained with an initial guess for 10,000 steps and fine-tuned for 20,000 steps.

Figure 16 presents the performance metrics of both methods across different noise levels in the Fisher-KPP problem, with each noise level tested over five random trials. In terms of PDE parameter accuracy, BiLO consistently outperforms PINN across varying values of $w_{\text{dat}}$ and noise levels. Notably, the optimal $w_{\text{dat}}$ for PINN depends heavily on the noise level. For example, a relatively large $w_{\text{dat}} = 10$ works well for low noise ($\sigma^2 = 10^{-4}$) but performs poorly at higher noise levels ($\sigma^2 = 10^{-2}$), suggesting that selecting the optimal $w_{\text{dat}}$ in practice may be challenging. In contrast, BiLO is more robust to noise and maintains consistent performance across all noise levels.

When evaluating the accuracy of the neural network solution $|u_{NN} - u_{FDM}|_\infty$, BiLO consistently delivers accurate solutions regardless of the noise level. But for PINN, larger $w_{\text{dat}}$ leads to less accurate solutions. Considering the data loss $\mathcal{L}_{data}$, the metric should ideally be approximately equal to the variance of the noise: a smaller value indicates overfitting, while a larger value suggests underfitting. The data loss of BILO is close to the noise level. In contrast, for PINN, smaller $w_{\text{dat}}$ leads to underfitting, while larger $w_{\text{dat}}$ leads to overfitting.

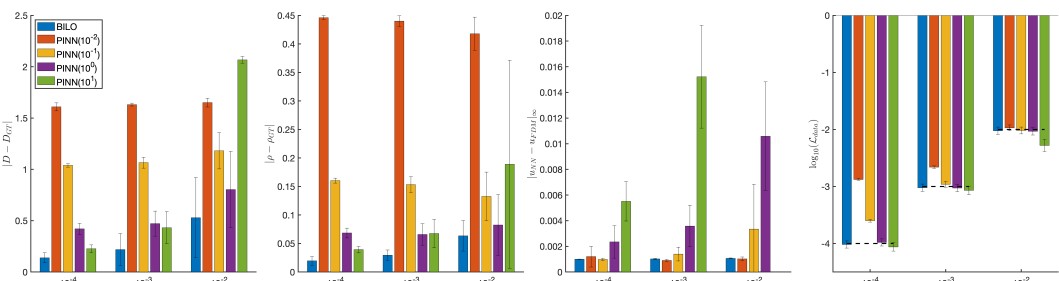

Figure 16: Comparison of performance metrics across different methods for varying variance ($10^{-4}$, $10^{-3}$, $10^{-2}$ as the x-axis) of the noise in the Fisher-KPP problem. Each subplot corresponds to a specific metric: (a) $|D - D_{GT}|$, the absolute error in $D$; (b) $|\rho - \rho_{GT}|$, the absolute error in $\rho$; (c) $|u_{NN} - u_{FDM}|_\infty$, the infinity norm error of predicted $u$; and (d) $\log_{10}(\mathcal{L}_{data})$, the logarithm of the data loss. The bars represent the mean values with error bars denoting the standard deviation for each method. The methods include BILO and PINN with varying $w_{\text{dat}}$ ($10^{-2}$, $10^{-1}$, $10^0$, $10^1$). The dashed lines in subplot (d) indicate the variance of the noise. A smaller data loss compared to the noise indicates a tendency to overfit the data, while a larger data loss compared to the noise indicates underfitting.

