# OpenReview forum: "BiLO: Bilevel Local Operator Learning for PDE inverse problems"
_ICLR.cc/2025/Conference — Submitted to ICLR 2025_

### Official Review · Reviewer_HQjM · 2024-10-26

**Soundness:** 3
**Presentation:** 4
**Contribution:** 3
**Rating:** 6
**Confidence:** 5

**Summary:**

This work aims to infer parameters in parametric PDEs using some observed solution data. This type of problem is widely found in scientific and engineering applications. The authors propose a neural network method for solving inverse problems governed by partial differential equations (PDEs). This method contains two steps: the pre-train step using the data and collocation points and the fine-tune step using the automatic differentiation. The proposed approach incorporates the traits of PINN, operator learning, and the adjoint method.

**Strengths:**

1. This paper is well-written.
2. The proposed method is mesh-free, avoiding the difficulties caused by mesh generation.
3. The algorithm is easy to implement and can provide fast inference for inverse problems once it has finished training.

**Weaknesses:**

1. The PDE-constrained optimization problem considered in this work only involves the equality constraint, but in practice, the inequality constraints are typical, e.g., the box constraint.
2. The pre-train step is crucial, and this step is minimizing the residual loss with collocation points. When the solution has a low-regularity property, this pre-train step will have a large generalization error if the collocation points are not properly chosen. This will overshadow the capability of the proposed method.

**Questions:**

1. The initial parameter $\Theta$ seems important to the propsoed method. How do you choose it?
2. Again, I wonder how the proposed method works if the initial guess of $\Theta$ is far away from the ground truth.
3. In this work, the loss function is easy to construct since the equality constraint only exists. How do you generalize your method to inequality constraints? The inequality constraints are common in practice.
4. Which grids does the observed data locate? Please clarify this issue in the numerical experiments.
5. In this work, the noise is set to the Gaussian with a small variance. How about the noise with a large magnitude?
6. About the training procedure. The simultaneous gradient descent is applied in the fine-tuning step instead of the alternative gradient descent. Could you plot the training loss curve for this step?

---

> ### Author Response · Authors · 2024-11-14
> **Response to reviewer HQjM (Part 1)**
>
> We thank the reviewer for their detailed feedback and for highlighting the strengths of our work. We also appreciate the reviewer’s insightful questions. We address some concerns and questions below.
>
> > The PDE-constrained optimization problem considered in this work only involves the equality constraint, but in practice, the inequality constraints are typical, e.g., the box constraint.
>
> > In this work, the loss function is easy to construct since the equality constraint only exists. How do you generalize your method to inequality constraints? The inequality constraints are common in practice.
>
> We thank the reviewer for this interesting question. Our current focus is on handling PDE constraints, but we are very interested in expanding our method to address more complex constraints, including inequality constraints. We believe our approach could be combined with other techniques to handle these additional constraints effectively, and we would welcome any suggestions on relevant problems where this extension could be applied.
>
> For example, if we consider PDE-constrained optimization problem with inequality constraints on the PDE solution $u$ and constrain on the PDE parameter $\Theta\in \mathbb{R}$
> $$
> \begin{aligned}
> \min_{\Theta} & \sum_i(u(x_i)-\hat{u}_i)^2 \\\\
> \text{s.t.} \quad & c[u] \leq 0, \quad\\\\
> & \Theta \leq a \\\\
> & F(D^ku(x),...,u(x),\Theta) = 0\\\\
> \end{aligned}
> $$
> where $c$ is a functional of the PDE solution $u$.
>
> The state and box constraints could be handled by a penalty method or an augmented Lagrangian approach, while the PDE constraint is addressed through our method. If we use the penalty method, we have the following bilevel optimization problem:
>
> $$
> \min_{\Theta} \sum_i (u(x_i,\Theta;W) - \hat{u_i})^2  + \lambda g(c[u(\cdot,\Theta;W)]) + \mu g(a-\Theta)
> $$
>
> $$
> \text{s.t.} \quad  W^* = \arg \min_W \mathcal{L}_{\rm LO}(\Theta;W)
> $$
>
> where $\mathcal{L}_{LO}$ is our local operator loss, and exterior penalty function $g(z) = \max(0,z)^2$
> is used to handle the inequality constraints. The penalty parameter $\lambda$ and $\mu$ could evolve during the upper level optimization.
> We can rely on the lower level problem to approximate the local PDE operator, and other optimization techniques to control the descent direction of the parameters at the upper level.
>
> > Which grids does the observed data locate? Please clarify this issue in the numerical experiments.
>
> The training details are shown in Appendix D.
> For example, for the Fisher-KPP equation example (Section 3.1), the domain is $(x,t) \in [0,1]\times [0,1]$. The residual loss is evaluate on 51-by-51 grid points. The observation data is provided at 11 grid points at the final time $t=1$.
>
> > In this work, the noise is set to the Gaussian with a small variance. How about the noise with a large magnitude?
>
> While variance does not seems large, we generally choose a relatively small number of observation points.
> The sparse observation make it easier for the neural network to overfit the data.
> For example, for the Fisher-KPP equation example (Section 3.1), the domain is $(x,t)\in [0,1]\times [0,1]$. The observation data is provided at 11 grid points **at the final time $t=1$**.
>
> In the example from Section 3.2 (inferring the diffusion coefficient), this setup is a classical ill-posed inverse problem, where small noise in the observed $u$ can lead to large variations in $D$. Regularization is needed to prevent overfitting $D$. In such cases, both the form and the strength of the regularization are important to achieve good inference, although they are not the focus of this work. In Appendix E.2, Table 4, we cross-validate our results with different regularization strength, and we compare BILO with the adjoint method and PINNs with different data weight ($w_{dat}$)
>
> For our tumor inverse problem (Section 3.3), the segmentation data is noisy, and the model can be misspecified, as a simple PDE model does not explain all the complex phenomena of tumor growth. Our results have an accurate PDE solution without balancing between fitting data and solving the PDE (see Table 3 and Figure 4).
>
> > About the training procedure. The simultaneous gradient descent is applied in the fine-tuning step instead of the alternative gradient descent. Could you plot the training loss curve for this step?
>
> We have added a new figure in Appendix D (Figure 6) showing the training history of the unweighted losses (residual loss, residual-gradient loss, and data loss) during the fine-tuning stage of BiLO for the inverse problems with the Fisher-KPP equation (Section 3.1) and the Poisson equation with a variable diffusion coefficient (Section 3.2). The plot demonstrates that during parameter inference, as the PDE parameters change, the residual and residual-gradient losses remain stable while the data loss decreases. This indicates that the neural network continues to serve as an approximation of the local PDE solution operator as the PDE parameters evolve.

---

> ### Author Response · Authors · 2024-11-14
> **Response to reviewer HQjM (Part 2)**
>
> > The pre-train step is crucial, and this step is minimizing the residual loss with collocation points. When the solution has a low-regularity property, this pre-train step will have a large generalization error if the collocation points are not properly chosen. This will overshadow the capability of the proposed method.
>
> We agree that the pre-train step and the choice of collocation points is crucial.
> Before solving PDE-constraint optimization problem (the inverse problem), we need to have a good solver for the PDE (forward problem).
> In our work, we rely on minimizing the residual loss to solve the PDE.
> Therefore, as in PINN, the choice of collocation points is crucial.
> In other words, if we can not make a good approximation of the local PDE operator, the descent direction in the upper level will be bad.
>
> Our framework benefits from ongoing advances in the field of solving PDEs using neural network. Improvements in neural network architectures, training strategies and sampling strategies (discussed in Section 2.4), should be able to enhance BiLO’s performance for challenging PDEs as well. For example, methods such as random fourier features [1] or transformer architecture [2], are shown to improve PINNs for PDEs with oscillatory solutions or multi-scale phenomena. We believe that these techniques should benefit BiLO as well.
>
> In addition, our approach is not inherently tied to neural network representations of the solution, allowing flexibility in the choice of representation.
>
> [1] S. Wang, H. Wang, and P. Perdikaris, “On the eigenvector bias of Fourier feature networks: From regression to solving multi-scale PDEs with physics-informed neural networks,” Computer Methods in Applied Mechanics and Engineering, 2021.
>
> [2] Z. Zhao, X. Ding, and B. A. Prakash, “PINNsFormer: A Transformer-Based Framework For Physics-Informed Neural Networks,” arXiv, 2024.
>
> > The initial parameter seems important to the propsoed method. How do you choose it?
>
> We agree that an initial guess is important for many gradient-based optimization problems. For the initial guess of PDE parameters, we carefully chose examples to ensure they are non-trivial and interesting. For example, in all the examples for inferring unknown functions, the ground truth is non-smooth, while the initial guess is smooth.
>
> Here is a summary of the initial guess used in the paper.
>
> (1) Section 3.1 (Fisher-KPP equation), the relative deviation from the ground truth to the initial guess is $|D_{GT}-D_0|/D_0$ = 100% (same for $\rho$). In addition, while the solution of the PDE evolve from $t=0$ to $t=1$,
> the observation data is provided **only at final time t=1**.
> This is motivated by our tumor growth problems where only single-time snapshots are available.
> This setup introduces a narrow valley in the loss landscape along the diffusion coefficient $D$ (Figure 3), making it challenging to infer the diffusion coefficient accurately. If the observation data is provided at all time, the landscape would be more benign, resulting in a simpler problem.
>
> (2) Section 3.2 (variable diffusion coefficient), the initial guess is a constant function $D_0(x)=1$, while the ground truth is a  **non-smooth** hat function, with a kink at x=0.5.
> The relative deviation in the infinity norm is $\Vert D_{gt}-D_0\Vert_{\infty}/\Vert D_0\Vert_{\infty}$  = 25%.
>
> (3) For section 3.4 (tumor inverse problem), we use an established heuristic in the literature to estimate the parameter. The initial $D$ and $rho$ are chosen by approximating the complex brain geometry using a homogeneous sphere, **which ignore the complex geometry of the brain**.
>
> (4) Appendix F.1 (Inferring initial condition of Heat equation), the initial guess is a smooth $\sin$ function, while the ground truth is a **non-smooth** hat function.
>
> (5) Appendix F.2 (invisid burger's equation), the initial guess is a smooth function, and the ground truth is **discontinuous with a shock**.
>
> (6) Appendix F.3 (2D darcy flow), the initial guess and the ground truth are based on independent draws from a Gaussian random field, which looks very different (second row of Figure 11 and Figure 12). The ground truth is also **discontinuous** due to an extra thresholding operation. The setup of this problem follows PINO (Li et al., 2024)
>
> > Again, I wonder how the proposed method works if the initial guess of is far away from the ground truth.
>
> As with many gradient-based optimization methods, our approach is likely to converge to a nearby local minimum, and the results depends on the optimization algorithm used. In our work, we apply the Adam optimizer with a constant learning rate at both levels of optimization. It would be interesting to explore other optimization methods for this purpose. For instance, SGD might help to escape from local minima. Since the upper-level problem is relatively low-dimensional, a second-order optimization method such as L-BFGS could potentially enhance convergence. We leave this for future work.

---

> > ### Comment · Reviewer_HQjM · 2024-11-15
> >
> > Thanks for the authors response. Now I have no question.

---

> > > ### Author Response · Authors · 2024-11-24
> > >
> > > We thank the reviewer for the response. We took the initiative to conduct additional experiments to further address the question on “How about the noise with a large magnitude?” and to analyze the performance of BiLO under varying noise levels. These new results are detailed in Appendix I. We compared BiLO and PINN on the Fisher-KPP problem across different noise levels. As shown in Fig.16, BiLO consistently outperforms PINN (with various weights of data loss) in terms of the accuracy of the estimated PDE parameters and the accuracy of the PDE solution, and maintains robust performance across all noise levels. In contrast, PINN’s performance is highly sensitive to the choice of weight.
> > >
> > > We sincerely appreciate your feedback, which has directly contributed to strengthening this work. We hope these additional efforts and insights address any remaining concerns. We would greatly appreciate it if you could consider raising your score or provide further suggestions for improvement.

---

### Official Review · Reviewer_4nyT · 2024-10-27

**Soundness:** 3
**Presentation:** 2
**Contribution:** 3
**Rating:** 6
**Confidence:** 4

**Summary:**

This paper proposes BiLO, which formulates upper and lower inverse problems into bilevel optimization. BiLO integrates adjoint methods and PINNs, while eliminating the need for balancing residual and data loss. BiLO has high robustness to sparse and noisy data, and effectively infers unknown PDE parameters and functions through auxiliary variables. Experimental results show that BiLO enforces strong PDE constraints, is robust to sparse and noisy data, and eliminates the need to balance the residual and the data loss, which is inherent to the soft PDE constraints in many existing methods.

**Strengths:**

- **Elimination of Residual-Data-Loss Balancing**. By separating the optimization process into two levels, BiLO eliminates the need to balance the residual and data loss;
- **Local Operator Learning**. Local PDE solution operators ensure precise gradient computation and lead to faster parameter inference.
- **Robustness**. Experimental results show that BiLO can handle noisy measurements and unseen data.
- **Generalization Power**. The proposed method could infer unknown functions (e.g., variable diffusion coefficients) with the auxiliary variables.

**Weaknesses:**

- **Lack of Theoretical Guarantees**. The paper offers mainly empirical results.
- **Scalability Concerns**. The experiments are limited to low-dimensional PDE problems. It is unclear how the method scales to higher-dimensional or more complex PDEs.
- **Computational Overhead**. Although the method achieves accurate results, solving both upper and lower-level optimization problems simultaneously introduces computational complexity.
- **Presentation**. The presentation of this paper could be further improved. The paper also seems to be completed in a rush and needs further proofreading. There are multiple typos (e.g., line 37, "... or deep learning, methods.", redundant comma) and duplicate bibitems (e.g., second and third last references on pp.15 are identical).

**Questions:**

- On line 177, "The use of Lu0 is not mandatory for training the local operator with fixed Θ0, though it can speed up the training process", does it mean fixed initial parameters throughout training or just during initialization?
- Scalability. How does BiLO perform on higher-dimensional PDEs? How could BiLO reduce computational overhead for large-scale problems?
- Can the authors provide a more rigorous theoretical analysis of the convergence behavior for the bilevel optimization process?
- Comparison with SOTA Neural Operators. The paper focuses on comparing BiLO with PINNs. Could the authors provide some comparison over neural operators (e.g., DeepONet, FNO)?

---

> ### Author Response · Authors · 2024-11-14
> **Response on Theoretical Analysis and Scalability Concerns**
>
> We thank the reviewer for their detailed feedback and for highlighting the strengths of our work. We also appreciate the reviewer’s insights on areas for improvement. We address some concerns and questions below.
>
> > Lack of Theoretical Guarantees. The paper offers mainly empirical results.
>
> > Can the authors provide a more rigorous theoretical analysis of the convergence behavior for the bilevel optimization process?
>
> We acknowledge that our results are primarily empirical. However, we have established the consistency of our algorithm: under certain conditions, the approximate gradient we use is indeed the true gradient of the upper-level objective (see the proposition before Section 2.4 and Appendix C). While our theoretical framework currently relies on the maximum principle for a linear elliptic PDE, our numerical examples extend to nonlinear (Fisher-KPP), parabolic (heat equation), and hyperbolic PDEs (inviscid Burgers' equation), demonstrating the method’s broader applicability.
>
> We are actively working toward a more comprehensive theoretical understanding of our approach, which we recognize as challenging but promising. To the best of our knowledge, many existing methods for general bilevel optimization require the lower-level problem to satisfy certain convexity conditions for convergence guarantees. In our case, the lower-level problem—training a neural network to approximate the PDE solution operator—is inherently non-convex and high-dimensional. While this poses challenges, we believe that the structure of the PDE itself provides opportunities for efficient and convergent solutions, as highlighted in our initial proposition.
>
> > Scalability Concerns. The experiments are limited to low-dimensional PDE problems. It is unclear how the method scales to higher-dimensional or more complex PDEs.
>
> > Scalability. How does BiLO perform on higher-dimensional PDEs? How could BiLO reduce computational overhead for large-scale problems?
>
> We appreciate the reviewer’s interest in the scalability of our method. We acknowledge that solving high-dimensional or complex PDEs poses a significant challenge for most PDE-constrained optimization algorithms, not just BiLO. Nevertheless, our method has flexibility that positions it well to adapt to such settings, as outlined below:
>
> 1. High-dimensional PDEs: The BiLO framework is designed for general PDE-constrained optimization problems, assuming the underlying PDE can be solved accurately. Our approach is not inherently tied to neural network representations of the solution, allowing flexibility in the choice of representation. However, the neural network-based representation of solutions is a strength in high dimensions, as neural networks are generally less susceptible to the curse of dimensionality compared to traditional methods like finite element or finite difference methods.
>
> 2. Application to Real-World Problems: In our last example, we applied BiLO to a 2D inverse problem related to tumor growth, using real patient data and a nonlinear Fisher-KPP model. This example is a step towards scaling up BiLO, addressing the complexities introduced by noisy data and potential model misspecifications. We are currently working on extending this example to 3D, which will further demonstrate the method's scalability.
>
> 3. Potential for Further Improvements: Our framework benefits from ongoing advances in the field. Improvements in neural network architectures or training strategies from the PINN community [1], or innovations in numerical representations, should be able to enhance BiLO’s performance for high-dimensional problems as well.
>
> [1] Z. Hu, K. Shukla, G. E. Karniadakis, and K. Kawaguchi, “Tackling the curse of dimensionality with physics-informed neural networks,” Neural Networks, 2024.

---

> ### Author Response · Authors · 2024-11-14
> **Response on computational overhead and comparison with Neural Operators**
>
> > Computational Overhead. Although the method achieves accurate results, solving both upper and lower-level optimization problems simultaneously introduces computational complexity.
>
> Compared with PINN, the primary source of computational overhead in our method comes from calculating the derivative of the residual with respect to the PDE parameters, which requires higher-order derivatives compared to the residual alone. This trade-off reflects an accuracy-speed balance. While our method may have a higher per-step cost, it can potentially require fewer steps to converge due to more accurate gradient directions.
>
> In Appendix G, we provide details on the maximum allocated memory and the seconds-per-iteration for both PINN and BiLO. The computational overhead varies depending on factors such as the specific PDE and the number of PDE parameters involved. A more detailed analysis of the accuracy-speed trade-off is part of our future work.
>
>
> Additional training details for the numerical examples are presented in Appendix D.
> In Figure 5, we plot the training history of the parameter $D$ and $\rho$ for the Fisher-KPP example in Section 3.1.
> In Figure 7, we show the training history of metrics ($L_2$ and $L_\infty$ error to the ground truth) for the variable diffusion coefficient example in Section 3.2.
> In both cases, we observe that BiLO converges at least as quickly as PINN when an appropriate data weight is chosen and ultimately finds a better minimum. We attribute this to the improved descent direction for PDE parameters provided by BiLO (as discussed in Section 2.4 around Equation 13).
>
> > Presentation. The presentation of this paper could be further improved.
>
> We thank the reviewer for the suggestion. We correct the typos.
>
> > On line 177, "The use of Lu0 is not mandatory for training the local operator with fixed Θ0, though it can speed up the training process", does it mean fixed initial parameters throughout training or just during initialization?
>
> In the pretraining stage, our goal is to initialize the neural network as a local operator at a specific $\Theta_0$.
> We fix the PDE parameter $\Theta_0$ and train the local operator using the residual loss and the residual-gradient loss. This setup is similar to PINNs, as minimizing the residual loss here is equivalent to solving the PDE at at $\Theta_0$.
>
> If a numerical solution is available at $\Theta_0$, denoted as $u_0$, we can also include the data loss $L_{u0}$ to help with training, which generally accelerates convergence in the pretraining stage.
> This assumption is mild, as it requires only a single numerical solution, unlike neural operators, which typically need a large dataset of solutions.
>
> > Comparison with SOTA Neural Operators. The paper focuses on comparing BiLO with PINNs. Could the authors provide some comparison over neural operators (e.g., DeepONet, FNO)?
>
> We believe that BILO and neural operator (NO) are complementary, each with strength and weakness in different settings. We discuss this in Section 2.4 of our paper, and elaborate here for clarity.
> A NO learn the parameter-to-solution map. It is trained by a synthetic datasets that consist of many pairs of PDE parameters and their corresponding numerical solutions. Once trained, it is fast to evaluated. The vanilla version of NO does not make use of the PDE residual loss, and therefore it has limited ability to generalize to out-of-distribution parameters.
> We believe NOs are suitable: (1) when we have good **prior knowledge of distribution of the parameters**, and (2) in **many-query** problems, such as Bayesian inference for uncertainty quantification, where we need to solve the PDE for many different parameters.
>
> In contrast, BiLO serve as an **algorithm for PDE-constrained optimization problems** (similar to adjoint methods or PINNs), relying on the PDE residual loss and **not requiring synthetic datasets**. This makes it more suitable for **single-query** settings, where the goal is to find the best-fit parameters from a single observational dataset.
>
> We illustrate these differences in our numerical examples. In Section 3.1 (Fisher-KPP equation), Table 1, we compare BiLO with DeepONet, showing that NO performance depends on the sampling of the parameter space, with performance degradation if the ground truth is out of the training distribution.
>
> In Appendix E2.2 (inferring variable diffusion coefficient), we also compare BILO with DeepONet.
> To train a NOs that maps variable diffusion coefficient $D(x)$ to the PDE solution, the usual practice is to sample $D(x)$ from a Gaussian random field with a specified length scale and covariance function.
> We show that the performance of DeepONet depends on the length scale of the Gaussian random field in the training data.
> The details are provided in Appendix E (Comparison with Neural Operators).

---

> ### Comment · Reviewer_4nyT · 2024-11-26
>
> Thank you for your clarification. I am keeping my original rating.

---

### Official Review · Reviewer_mdLb · 2024-10-30

**Soundness:** 2
**Presentation:** 3
**Contribution:** 2
**Rating:** 6
**Confidence:** 4

**Summary:**

This paper intends to solve PDE constrained inverse problem with neural operators, which is formulated into a bi-level optimization problem. In the lower level optimization, a neural operator is learned and trained on a fixed initial guess of PDE parameter $\Theta_0$, which is named "local neural operator" by the authors. In the upper level optimization, the optimal or targeted parameter $\Theta$ is approximated.

In the experiment, the authors benchmarked their method on a tumor growth and imaging model and compared results to PINN-based method.

**Strengths:**

The paper basically argues for solving inverse problem with neural operators instead of PINN for both efficiency and accuracy. The support of the argument is mostly empirical. However, the experiment is well designed in both diversity and careful measurement, e.g., in addition to measuring accuracy of solution and unknow parameter (function), the author deliberately choose the initial guess that is far from ground truth in their experiment, which is a strong evidence for the effectiveness of the method.

**Weaknesses:**

What is the motivation of using neural operator for PDE-constrained inverse problem? Neural operator is known for its generalization on various input functions, boundary conditions, initial data, etc. However, in the setting of this paper, a "local" neural operator is learned, meaning narrow generalization ability and abandoning the advantage of neural operators. Could the authors elaborate on the advantages of using a local neural operator versus a more general neural operator in this context? What specific benefits does the local approach provide for PDE inverse problems that outweigh the loss of broader generalization ability?

Meanwhile, PINN is known for low requirement of data and being suitable for inverse problem. The experiment indeed shows that the improvement of the method proposed here over PINN is quite marginal considering noise (see Table 2&3). Given the relatively small improvement over PINN shown in Tables 2 and 3, could the authors provide a more comprehensive discussion of the advantages of their method? Are there specific scenarios where the proposed approach significantly outperforms PINN that may not be captured in the current experiments?

Also, how does the proposed method compare to the bi-level optimization approach for PINNs presented in [1]? Could the authors discuss the key differences and potential advantages of their method over this existing work?

[1] Hao, Zhongkai, et al. "Bi-level Physics-Informed Neural Networks for PDE Constrained Optimization using Broyden's Hypergradients." The Eleventh International Conference on Learning Representations.

**Questions:**

N.A.

---

> ### Author Response · Authors · 2024-11-13
> **Difference from PINN and NO**
>
> We thank the reviewer for their feedback and insightful questions. We will clarify the motivations and advantages of our approach.
>
> > What is the motivation of using neural operator for PDE-constrained inverse problem? ... Could the authors elaborate on the advantages of using a local neural operator versus a more general neural operator in this context? What specific benefits does the local approach provide for PDE inverse problems that outweigh the loss of broader generalization ability?
>
> We believe that BILO and neural operator (NO) are complementary, each with strength and weakness in different settings. We discuss this in Section 2.4 of our paper, and elaborate here for clarity.
> A NO learn the parameter-to-solution map. It is trained by a synthetic datasets that consist of many pairs of PDE parameters and their corresponding numerical solutions. Once trained, it is fast to evaluated. The vanilla version of NO does not make use of the residual of the PDE.
>
> However, we believe the generalization ability is limited to in-distribution generalization, and NOs have limited ability for out-of-distribution (OOD) generalization. For example, if a NO is trained on $\Theta \in [a,b]$ but the true parameter is $\Theta_0 \notin [a,b]$, the NO will not be able to give a good approximation of the solution at $\Theta_0$.
> For example, in Table 1, we show that solving inverse problems using NO depends the sampling of the parameter space, and the results degrades significantly if the true parameter is OOD.
> The sampling is more challenging if the unknown is a function. In Appendix E2.2, we infer the unknown variable diffusion coefficient of a poisson equation. Usually in NO, the samples are generated from some Gaussian random field with certain length scale and covariance structure. In Figure 6, we show that the performance of NO depends on the length scale of the gaussian random field of the synthetic dataset.
>
> We believe NOs are suitable: (1) when we have good **prior knowledge of distribution of the parameters**, and (2) in **many-query** problems, such as Bayesian inference for uncertainty quantification, where we need to solve the PDE for many different parameters.
>
> We do note that there are strategies to handle OOD in NO. The physics-informed version of NOs make use of the PDE residual for "instance-wise fine-tuning" when the true parameter is OOD. However, this would introduce additional trade-off: fine-tune the NO for a particular parameter might degrade the performance overall on other parameters.
>
> BILO, on the other hand, is more akin to an **algorithm for PDE-constrained problems**, similar to adjoint methods or PINNs. It's more suitable in **single-query** settings, where our objective is to find the best-fit parameters based on one set of observational data, and **synthetic datasets are not needed**.
>
> > Meanwhile, PINN is known for low requirement of data and being suitable for inverse problem....could the authors provide a more comprehensive discussion of the advantages of their method? Are there specific scenarios where the proposed approach significantly outperforms PINN that may not be captured in the current experiments?
>
> We discuss the difference between BILO and PINN in Section 2.4 of our paper, and elaborate here for clarity. We agree that PINN is can achieve good results for solving inverse problems -- **if we can find the right balance between the data loss and the residual loss**. Our experience is that finding the right balance can be challenging, as the trade-off is inherent from the unconstrained optimization problem.
> The optimal weights depend on the problems and the noise level. For example, in Table 1,2,3 and 4, the optimal $w_{dat}$ can be different.
> The same concern about the trade-off leads to the work of [1], which aim to separate the data loss and the residual loss.

---

> ### Author Response · Authors · 2024-11-13
> **Difference from BI-PINN**
>
> > Also, how does the proposed method compare to the bi-level optimization approach for PINNs presented in [1]? Could the authors discuss the key differences and potential advantages of their method over this existing work?
>
> We thank the reviewer for pointing out the work in [1], which provides an interesting perspective on bi-level optimization for PDE-constrained problems. While our approach and the one presented in [1] share a focus on PDE-constrained optimization, they are designed with distinct structural differences. We discuss these distinctions in the revised manuscript and summarize them here for the reviewer’s convenience.
>
> In PINN and [1], the PDE solution is represented by a neural network $u(x,W)$ (where $W$ is the network weight). **Notice that $\Theta$ is not an input to the neural network**. Therefore, in both PINN and [1], the data loss is given by
>
> $$L_{data}(W) = \frac{1}{N} \sum_i(u(x_i;W) - \hat{u}_i)^2$$
>
> In PINN and [1], the PDE constraint becomes an optimization problem:
>
> $$L_{res}(W,\Theta) = \frac{1}{N} \sum_i F(D^ku(x_i;W),...,u(x_i;W),\Theta)^2.$$
>
> PINN takes a penalty-like approach and solve the following optimization problem:
> $$\min_{W,\Theta}  L_{res}(W,\Theta) + w_{dat} L_{data}(W).$$
>
> In [1], the residual loss is separate from the data loss, and the bilevel optimization problem is
> $$
> \begin{aligned}
> \min_{\Theta}  L_{data}(W^*(\Theta)) \\
> \text{s.t.} \quad W^*(\Theta) = \arg \min_W L_{res}(W,\Theta).
> \end{aligned}
> $$
>
> Notice that **$L_{data}$ does not depends on $\Theta$ directly**. The dependence is indirect through the minimizer of lower level problem. The descent direction of the PDE parameter is given by
> $$\nabla_\Theta L_{data}(W^*(\Theta)) = \nabla_W L_{data}(W^*(\Theta)) \nabla_\Theta W^*(\Theta),$$
> where the hypergradient $\nabla_\Theta W^*(\Theta)$ is the focus of many bilevel optimization algorithm:
> $$\nabla_\Theta W^*(\Theta) = - \left[\frac{\partial^2 L_{res}}{\partial W \partial W^T}\right]^{-1} \cdot \frac{\partial^2 L_{res}}{\partial W \partial \Theta^T}.$$
> In [1], Broyden's method is used to compute the hyper-gradient, which is based on the low-rank approximation of the inverse Hessian.
>
> Our approach has three key differences to [1]:
>
> 1. Firstly, **$\Theta$ is also an input to our neural network**. Our neural network is $u(x,\Theta; W)$. Under this representation, the data loss depends on the PDE parameters directly:
> $$
> L_{data}(W,\Theta) = \frac{1}{N} \sum_i(u(x_i ,\Theta ;W) - \hat{u}_i)^2.
> $$
>
> 2. It seems that by including $\Theta$ as an input to the neural network, we can compute $\nabla_\Theta L_{data}$ directly. However, additional conditions are needed to ensure that this direction is actually a descent direction. This leads to the second difference: $u(x,\Theta;W)$ should be the **local PDE operator at $\Theta$**. It's not enough to just solve the PDE at $\Theta$. We also require that the gradient of the PDE residual with respect to $\Theta$ to be 0. This leads to our definition of residual-gradient loss (eq.7). Therefore, our lower level problem is different from [1]:
> $$
> \begin{aligned}
> &\min_{\Theta}  L_{data}(W^*(\Theta),\Theta) \\
> \text{s.t.} \quad W^*(\Theta) = \arg \min_W L_{res}(W,\Theta) + L_{rgrad}(W,\Theta)
> \end{aligned}
> $$
>
> 3. Thirdly, we solve the optimization problem by **simultaneous gradient descent**, which is easy to implement. We don't need to explicitly compute the hypergradient.
>
> Direct comparison with [1] remain a future work at this point.
>
> We add the following paragraph to the revised manuscript before Section 3.
>
> "The challenge of balancing trade-offs also motivated [1], which applies a bilevel optimization framework to PDE inverse problems by representing the PDE solution with a neural network, using the residual loss for the lower-level problem, and approximating the upper-level hypergradient with Broyden's method. In contrast, our approach incorporates the PDE parameter as part of the network input, with the lower-level problem focused on approximating the local operator, allowing more direct computation of the upper-level descent direction."

---

> ### Comment · Reviewer_mdLb · 2024-11-18
>
> > **Notice that $\Theta$ is not an input to the neural network.**
>
> In fact, in the original paper of PINN [1], in Section 4. "Data-driven discovery of partial differential equations", parameters of PDE $\Theta$ can be learned from data, which is one of the advantageous feature of PINN. Therefore, the parameters of PDE $\Theta$ can be optional learnable parameter in the framework of PINN, and it is not an improvement of the proposed method.
>
> Additionally, since [2] is directly related to your work, comparison with this method is vital to justify any advantage of the proposed method. Though there exists difference between the two methods and hence potential improvement, current work lacks the important comparison. Therefore, my score remains.
>
> [1] Raissi, Maziar, Paris Perdikaris, and George E. Karniadakis. "Physics-informed neural networks: A deep learning framework for solving forward and inverse problems involving nonlinear partial differential equations." Journal of Computational physics 378 (2019): 686-707.
>
> [2] Hao, Zhongkai, et al. "Bi-level Physics-Informed Neural Networks for PDE Constrained Optimization using Broyden's Hypergradients." The Eleventh International Conference on Learning Representations.

---

> > ### Author Response · Authors · 2024-11-24
> >
> > Regarding the learnability of PDE parameters within the PINN framework, we fully agree PINN can also solve the inverse problem of learning PDE parameters from data. However, the primary contribution of our work lies not in demonstrating that PDE parameters are learnable but in proposing a new and efficient approach—BiLO—for solving the inverse problem, and we have made extensive comparison with PINN.
> >
> > In response to the reviewer’s request for comparison with BPN[2], we have conducted new experiments and included the results in appendix H. Specifically, we compare BiLO and BPN using the same problem setup as in [2], ensuring a fair and consistent evaluation. Our findings demonstrate the advantages of BiLO in terms of efficiency and performance. As shown in Appendix H Fig.15, BiLO achieves a parameter error below 0.01 in fewer than 80 lower-level iterations and just 6.4 seconds, compared to BPN, which requires 27 upper iterations (1728 lower-level iterations) and 231 seconds to achieve the same accuracy. This improvement is achieved through simultaneous gradient descent at both levels, leveraging the structure of the PDE constraints for efficiency, as detailed in the appendix.
> >
> > We appreciate the reviewer’s feedback, which has strengthened our work, and would be grateful if you could consider raising your score. If there are further concerns or areas for improvement, we would be happy to address them.

---

> > > ### Comment · Reviewer_mdLb · 2024-11-25
> > >
> > > Thanks for the additional comparison. The result is clear that emperically the proposed method is more efficient at solving inverse problem than PINN-based method. Now I am more convinced about the advantage of solving inverse problem with neural operators in both efficiency and accuracy. I also find the discussion of "Computational Cost" in Section G. of appendix helpful. Therefore, I change my score to 6.

---

### Official Review · Reviewer_KNrc · 2024-11-01

**Soundness:** 2
**Presentation:** 3
**Contribution:** 2
**Rating:** 6
**Confidence:** 4

**Summary:**

The paper investigates a method for parameter identification in PDEs (similar to optimal control). The idea is to replace the PDE constraint with novel functional that penalises both the residual as well as the derivative of the residual with respect to the parameters. The latter is local in the parameters, hence the motivation for the title of the paper. The paper also models the solution as a neural network similar to PINNS. This approach is equipped with an efficient learning strategy and initialisation. It is evaluated on various simple PDEs as well as a more complicated one related to tumour growth.

**Strengths:**

The problem considered is important and challenging. Across various communities do people search for efficient solutions, thus making the paper potentially significant. The paper is original in that the idea has not been considered before. The presentation is clear and the ideas do come across well. I really like the tumour growth example applying such methods to more than just box-standard applications.

**Weaknesses:**

The paper is based on a lot of heuristics and I have doubts that this idea in general will be useful for others solving similar problems.

1. The entire idea of local solutions is neither well-defined nor do I believe that what is being computed are local solutions in the sense that they actually solve the PDE in a neighbourhood of the parameters. Also they still use the global residual encouraging both local and global behaviour. It is not clear how to entangle the two: are both important?

2. The optimisation algorithm in general will not solve the bilevel optimisation problem. This problem has been studied for decades (or more; see the papers cited in the paper itself) and the community agrees that the proposed algorithm does not do it (in general). I agree with the arguments in the appendix but this is for a much simpler case and in my opinion will not generalise. This is why methods like the "adjoint method" etc exist. Note that this is also a current research topic in the optimisation community which investigate similar but still very different algorithms compared to the one proposed here. The paper also makes the claim that the PDE constraint "does not need to be solved to solved to optimality". I agree this is true but the statement needs to quantitative. There will be an accuracy that is needed to make sure the computed directions actually descent. This is completely ignored at present. See e.g.
Pedregosa, Fabian. "Hyperparameter optimization with approximate gradient." International conference on machine learning. PMLR, 2016.
The paper starts with a very good initial estimation of the solution which is probably the key reason why it works well in the numerical examples. I don't believe this generalises well to other problems.

Minor comments:
- line 199, typo a(x, f(x))

**Questions:**

1. Discuss (perhaps using numerical evidence) that indeed the solutions computed are solutions to the PDE. Also discuss (again potentially with numerical evidence) that the solutions computed are local solutions and not global solutions.
2. Discuss (perhaps using numerical evidence) what the proposed algorithm actually converges to stationary solutions (or even global minima) of the bilevel problem.

---

> ### Author Response · Authors · 2024-11-13
> **Clarification on "local PDE operator" and the scope**
>
> We thank the reviewer for the positive feedback and suggestions.
>
> > The entire idea of local solutions is neither well-defined nor do I believe that what is being computed are local solutions in the sense that they actually solve the PDE in a neighbourhood of the parameters.
>
>
> If $u(x,\theta)$ is the full PDE solution operator, then $F(D^ku(x,\theta)...u(x,\theta))=0$ for all $\theta$. This implies that for a fixed $\Theta_0$, $F(D^ku(x,\Theta_0)...u(x,\Theta_0))=0$ and $\nabla_\theta F(D^ku(x,\Theta_0)...u(x,\Theta_0))=0$. Our definition of a local PDE operator resembles a first-order condition, or a necessary condition, for a full PDE solution operator.
>
> The local operator aims to provide an approximation rather than a precise solution in the neighborhood of the parameter $\Theta_0$.
> By Taylor expansion,
> $$F(D^ku(x,\Theta_0+\delta)...u(x,\Theta_0+\delta))= F(D^ku(x,\Theta_0)...u(x,\Theta_0)) + \delta\nabla_\theta F(D^ku(x,\Theta_0)...u(x,\Theta_0)) + O(\delta^2)$$
> our condition implies that both the leading and first-order terms are zero, so the residual of the local operator at $\Theta_0+\delta$ is $O(\delta^2)$.
>
> As shown in Figure 1 and Figure 3, evaluating our trained local operator at a nearby parameter $\Theta_0+\delta$ gives a good approximation of the PDE solution at $\Theta_0+\delta$.
>
> If the PDE solution depends smoothly on the parameter, the local PDE operator will exist. In our numerical experiments, such as inferring the initial condition of an inviscid Burgers equation (Appendix F.2), where the solution does not depend smoothly on the initial condition, our method still performs well, suggesting that the condition could be relaxed. Developing a theoretical understanding of the local PDE operator's properties is part of our future work.
>
> > Also they still use the global residual encouraging both local and global behaviour. It is not clear how to entangle the two: are both important?
>
> For both the neural operator and our local operator, the function of interest has the form $u(x,\Theta)$. $x\in \Omega$ is the spatial-temporal variable, where $\Omega$ is the domain of PDE. $\Theta \in I$ is the PDE parameter, where I is the range of the parameter.
>
> To ensure that the PDE is solved, the collocation point for computing the residual $x_i$ are sampled "global" in $\Omega$.
>
> For neural operators (FNO, DeepONet), training also involves sampling $\Theta_i$ from the parameter space $I$, and the corresponding numerical solutions are used to train the neural operator.
>
> In BILO, by contrast, we do not need to sample $\Theta_i$ from $I$. We are akin to the adjoint method. We start with an initial guess $\Theta^(0)$ and iteratively update the parameter to minimize the data loss.
>
> > The optimisation algorithm in general will not solve the bilevel optimisation problem. This problem has been studied for decades (or more; see the papers cited in the paper itself) and the community agrees that the proposed algorithm does not do it (in general)....
>
> We would like to clarify the scope of our work: our method is designed for PDE-constrained optimization problem, and is not intended for general bilevel optimization problem.
>
> We agree that our algorithm does not solve a general bilevel optimization problem. Our focus is on PDE-constrained optimization problems. In our case, the lower-level optimization problem arises specifically from the PDE constraint, giving it additional structure compared to a more general optimization problem.
>
>
> > line 199, typo a(x, f(x))
> We appreciate the reviewer’s careful attention to detail. This is actually not a typo, but we realize it could be clearer. We will revise the text to clarify.
>
> By our definition, $a(x, z) := F(D^ku(x,z),...,D u(x,z), u(x,z), z)$. Then $a(x, f(x)) = 0$ implies that $F(D^ku(x,f(x)),...,D u(x,f(x)), u(x,f(x)), f(x))$ meaning that $u(x,f(x))$ is a solution to the PDE, which has 0 residual.
>
> > Discuss (perhaps using numerical evidence) that indeed the solutions computed are solutions to the PDE. Also discuss (again potentially with numerical evidence) that the solutions computed are local solutions and not global solutions.
>
> We show what happens if we evaluated our trained local operator at nearby parameters in Figure 1 and Figure 3.
> While our local operator is trained only at $\Theta_0$, evaluating our local operator at a nearby parameter $\Theta_0+\delta$ gives a good approximation to the PDE solution at $\Theta_0+\delta$.
>
> > Discuss (perhaps using numerical evidence) what the proposed algorithm actually converges to stationary solutions (or even global minima) of the bilevel problem.
>
> Our method is designed for the PDE-constrained optimization problem. In this case, the minimum is supposed to be the ground truth parameters and their corresponding PDE solution. In the numerical experiments. We show that our method can reliably recover the ground truth for many different PDE under noisy observations.

---

> > ### Comment · Reviewer_KNrc · 2024-11-15
> > **PDE-constrained optimisation**
> >
> > > > The optimisation algorithm in general will not solve the bilevel optimisation problem. This problem has been studied for decades (or more; see the papers cited in the paper itself) and the community agrees that the proposed algorithm does not do it (in general)....
> > >
> > > We would like to clarify the scope of our work: our method is designed for PDE-constrained optimization problem, and is not intended for general bilevel optimization problem.
> > >
> > > We agree that our algorithm does not solve a general bilevel optimization problem. Our focus is on PDE-constrained optimization problems. In our case, the lower-level optimization problem arises specifically from the PDE constraint, giving it additional structure compared to a more general optimization problem.
> >
> > Many thanks for the reply. Although when discussing the related work I was talking about generic bilevel optimisation, here I was referring to the specific problem of PDE-constrained optimisation. The authors may want to reconsider their reply given that it was a miscommunication.

---

> > > ### Author Response · Authors · 2024-11-25
> > >
> > > We thank the reviewer for the clarification and for carefully going through our argument in the appendix. We agree that ensuring the descent direction’s accuracy is important and that the interplay between the dynamics of the upper- and lower-level optimizations needs further investigation. We also appreciate the reference to Pedregosa’s work, which, while not directly applicable to our method, provides a valuable perspective on studying convergence of bilevel optimization algorithms. We have cited this work in the updated manuscript to acknowledge its relevance. Indeed, as a next step, we aim to conduct a more refined analysis, such as bounding the error in the descent direction based on the error in the lower level problem.
> > >
> > > In this work, we conducted experiments on a variety of problems, including nonlinear (Fisher-KPP), parabolic (heat equation), hyperbolic (inviscid Burgers’ equation), elliptic (2D Poisson), as well as cases with unknown scalar parameters, unknown functions, and tumor growth modeling using patient data. We hope these experiments help alleviate some concerns regarding the generalizability of our method. Additionally, our choice of hyperparameters (such as the learning rates) is generic and not specifically tailored to each problem. Our algorithm can be modified to incorporate a tolerance term for the lower-level problem—e.g., performing upper-level optimization only when the lower-level loss is below a certain threshold. While we observed that this approach produces a deterministic trajectory as the adjoint method (as shown by the red dashed line in Fig. 1(b)), it comes at the cost of slower performance compared to simultaneous gradient descent, which we believe offers a more practical cost-accuracy trade-off for many applications.
> > >
> > > We appreciate the reviewer’s insightful feedback, which has helped strengthen our work and will guide our future research.

---

> > > > ### Comment · Reviewer_KNrc · 2024-11-27
> > > >
> > > > I thank the authors for the thorough revision of their work. I agree with most of their replies. However, they don't change my overall reception of the paper. I think this is decent work that can be published at ICLR but also clearly has weaknesses and shortcomings.

---

> ### Author Response · Authors · 2024-11-13
> **clarification on the initial guess**
>
> > The paper starts with a very good initial estimation of the solution which is probably the key reason why it works well in the numerical examples. I don't believe this generalises well to other problems.
>
> We agree that an initial guess is important for many gradient-based optimization problems. For the initial guess of PDE parameters, we carefully chose examples to ensure they are non-trivial and interesting. For example, in all the examples for inferring unknown functions, the ground truth is non-smooth, while the initial guess is smooth.
>
> Our work focuses on PDE-constrained optimization rather than general bilevel optimization. We demonstrated the effectiveness of our method across various types of PDEs — nonlinear (3.1, 3.3 Fisher-KPP), parabolic (F.1 heat), hyperbolic (F.2 inviscid burger), elliptic (3.2, F.3 Poisson), unknown scalar (3.1,3.3), unknown function (F.1, F.2, F.3), and tumor inverse problem using patient data (3.3).
>
> Here is a summary of the initial guess used in the paper.
>
> (1) Section 3.1 (Fisher-KPP equation), the relative deviation from the ground truth to the initial guess is $|D_{GT}-D_0|/D_0$ = 100% (same for $\rho$). In addition, while the solution of the PDE evolve from $t=0$ to $t=1$,
> the observation data is provided **only at final time t=1**.
> This is motivated by our tumor growth problems where only single-time snapshots are available.
> This setup introduces a narrow valley in the loss landscape along the diffusion coefficient $D$ (Figure 3), making it challenging to infer the diffusion coefficient accurately. If the observation data is provided at all time, the landscape would be more benign, resulting in a simpler problem.
>
> (2) Section 3.2 (variable diffusion coefficient), the initial guess is a constant function $D_0(x)=1$, while the ground truth is a  **non-smooth** hat function, with a kink at x=0.5.
> The relative deviation in the infinity norm is $\Vert D_{gt}-D_0\Vert_{\infty}/\Vert D_0\Vert_{\infty}$  = 25%.
>
> (3) For section 3.4 (tumor inverse problem), we use an established heuristic in the literature to estimate the parameter. The initial $D$ and $rho$ are chosen by approximating the complex brain geometry using a homogeneous sphere, **which ignore the complex geometry of the brain**.
>
> (3) Appendix F.1 (Inferring initial condition of Heat equation), the initial guess is a smooth $\sin$ function, while the ground truth is a **non-smooth** hat function.
>
> (4) Appendix F.2 (invisid burger's equation), the initial guess is a smooth function, and the ground truth is **discontinuous with a shock**.
>
> (5) Appendix F.3 (2D darcy flow), the initial guess and the ground truth are based on independent draws from a Gaussian random field, which looks very different (second row of Figure 11 and Figure 12). The ground truth is also **discontinuous** due to an extra thresholding operation. The setup of this problem follows PINO (Li et al., 2024)

---

> ### Comment · Reviewer_KNrc · 2024-11-15
> **Local and global solutions**
>
> > > Discuss (perhaps using numerical evidence) that indeed the solutions computed are solutions to the PDE. Also discuss (again potentially with numerical evidence) that the solutions computed are local solutions and not global solutions.
> >
> > We show what happens if we evaluated our trained local operator at nearby parameters in Figure 1 and Figure 3. While our local operator is trained only at, evaluating our local operator at a nearby parameter gives a good approximation to the PDE solution at.
>
> This is just a repetition of the arguments in the paper that I do not agree with. It is debatable that Fig 1 and 3 show "solutions" in a concrete sense. They certainly show that solutions are roughly approximated.
>
> > > Discuss (perhaps using numerical evidence) what the proposed algorithm actually converges to stationary solutions (or even global minima) of the bilevel problem.
> >
> > Our method is designed for the PDE-constrained optimization problem. In this case, the minimum is supposed to be the ground truth parameters and their corresponding PDE solution. In the numerical experiments. We show that our method can reliably recover the ground truth for many different PDE under noisy observations.
>
> The answer as above applies here. Moreover, with noisy observations, why would the "ground truth parameters" be actually global solutions to the PDE-constrained optimisation problem?

---

> ### Author Response · Authors · 2024-11-19
>
> > This is just a repetition of the arguments in the paper that I do not agree with. It is debatable that Fig 1 and 3 show "solutions" in a concrete sense. They certainly show that solutions are roughly approximated.
>
> We appreciate the reviewer's careful examination of our figures and their emphasis on the distinction between "approximate" and "concrete" solutions. This feedback highlights an important nuance in interpreting our results.
>
> Let $u(x,\Theta_0;W)$ be our local PDE operator at $\Theta_0$. The residual of $u$ at $\Theta_0+\delta$ is given by:
>
> $$F(D^ku(x,\Theta_0+\delta)...u(x,\Theta_0+\delta))= F(D^ku(x,\Theta_0)...u(x,\Theta_0)) + \delta\nabla_\theta F(D^ku(x,\Theta_0)...u(x,\Theta_0)) + O(\delta^2)$$
>
> If we solve our lower level optimization problem to a small tolerance, say $\epsilon$, then the residual of the local operator at $\Theta_0+\delta$ is $O(\epsilon)$.
>
> Therefore, by "approximate solution", we mean the residual of $u$ at $\Theta_0+\delta$ is small.
>
> It seems the reviewer's concern is that, if a function (represented by a neural network) has small PDE residual at some discrete points, does it implies that the function is close to the PDE solution?
>
> We believe this is a challenging question on the theoretical foundation of PINN and remains an active research area.
> For example, the convergence of PINN is established for certain classes of PDEs under specific conditions (e.g., well-trained networks, sufficient collocation points, and PDEs with certain regularity condition) [1,2,3]
>
> [1] S. Mishra and R. Molinaro, “Estimates on the generalization error of Physics Informed Neural Networks (PINNs) for approximating PDEs,” 2023, arXiv:2006.16144.
>
> [2] Y. Shin, J. Darbon, and G. E. Karniadakis, “On the convergence of physics informed neural networks for linear second-order elliptic and parabolic type PDEs,” 2020, arXiv:2004.01806.
>
> [3] N. Doumèche, G. Biau, and C. Boyer, “Convergence and error analysis of PINNs,” 2023, arXiv:2305.01240.
>
>
> > The answer as above applies here. Moreover, with noisy observations, why would the "ground truth parameters" be actually global solutions to the PDE-constrained optimisation problem?
>
> This is an excellent question, and we appreciate the opportunity to further elaborate. This is why we choose to evaluate our method on different instance of noise and report the mean and standard deviation (Table 1, 2 and 4)
>
> To be more specific, suppose the ground truth parameters are $\Theta_{GT}$ and the corresponding PDE solution is $u_{GT}$. The noisy observation data is $\hat{u} = u_{GT} + \epsilon$, where $\epsilon$ is the noise. Then the optimal parameters $\Theta^*$ that minimizes the data loss is actually a random variable that depends on the noise, and most likely will not be the same as $\Theta_{GT}$. Searching for the globally optimal parameters $\Theta^*$ would be challenging, as most gradient-based optimization algorithm would only converge to a local minimum.
>
> However, the expectation of the optimal parameters $E[\Theta^*]$ should be close to $\Theta_{GT}$, and the expectation of the data loss $E[L(\Theta^*)]$ should be close to the expectation of the data loss of the ground truth $E[L(\Theta_{GT})]$, which is the variance of the noise. Therefore, we solve the inverse problem with difference instances of noise and report the mean and standard deviation of the error of the parameters and the data loss.

---

### Meta-Review · Area_Chair_ueat · 2024-12-21

**Metareview:**

This paper introduces a new approach for solving inverse problems for partial differential equations (PDEs). Specifically, the approach formulates the problem as a bilevel optimization problem. This efficient approach is also able to  enforce PDE constraints. Results demonstrate good performance on a range of different problems, including sparse and noisy data.

The reviewers agree that this work is novel and original. Moreover, the experiments are interesting. Also, the the paper is well written. However, there are also some concerns. Reviewers are skeptical how useful this framework will be in general. The experimental studies are limited to DeepONet. It would be good to see more experiments for different model architectures. There is lack of supporting theory for this approach, which is uncommon for optimization oriented ML papers.

While this is an overall nice paper, all reviewers indicated concerns in the final discussion. There is no strong advocate for this paper. I feel that the authors need to demonstrate that the proposed method is applicable more generally, either by providing a more extensive set of experiments that includes different architectures and a more diverse set of problems, or through more rigorous theory that supports the optimization approach. Thus, I recommend to reject this paper in its current form.

**Additional Comments On Reviewer Discussion:**

Authors used the rebuttal phase to respond the reviewer's concern. The responses where detailed, yet they didn't fully convince the reviewers to strongly advocate for this paper.

---

### Decision · Program_Chairs · 2025-01-22

Reject